# C3a Mediates Endothelial Barrier Disruption in Brain-Derived, but Not Retinal, Human Endothelial Cells

**DOI:** 10.3390/ijms252011240

**Published:** 2024-10-19

**Authors:** Hannah Nora Wolf, Larissa Guempelein, Juliane Schikora, Diana Pauly

**Affiliations:** Department of Experimental Ophthalmology, University Marburg, 35043 Marburg, Germany

**Keywords:** C3a, C5a, neuromyelitis optica spectrum disorder, vascular permeability, barrier integrity, human brain microvascular endothelial cells, human retinal microvascular endothelial cells

## Abstract

Neuromyelitis optica spectrum disorder (NMOSD) is associated with pathological aquaporin-4 immunoglobulin G (AQP4-IgG), which cause brain damage. However, the impact of AQP4-IgG on retinal tissue remains unclear. Additionally, dysregulated complement anaphylatoxins C3a and C5a, known to modulate the endothelial barrier, are implicated in NMOSD. This study evaluates the susceptibility of human brain microvascular endothelial cells (HBMEC) and human retinal endothelial cells (HREC) to C3a- and C5a-mediated stress using real-time cell barrier analysis, immunocytochemical staining, qPCR and IgG transmigration assays. The findings reveal that C3a induced a concentration-dependent paracellular barrier breakdown and increased transcellular permeability in HBMEC, while HREC maintained barrier integrity under the same conditions. C5a attenuated C3a-induced disruption in HBMEC, indicating a protective role. Anaphylatoxin treatment elevated transcript levels of complement component *C3* and increased C5 gene and protein expression in HREC, with no changes observed in HBMEC. In HBMEC, C5a treatment led to a transient upregulation of C3a receptor (*C3AR*) mRNA and an early decrease in C5a receptor 1 (*C5AR1*) protein detection. Conversely, HREC exhibited a late increase in C5aR1 protein levels. These results indicate that the retinal endothelial barrier is more stable under anaphylatoxin-induced stress compared to the brain, potentially offering better protection against paracellular AQP4-IgG transport.

## 1. Introduction

Hyperpermeability of the vascular endothelium is a hallmark of the aquaporin-4-autoantibody mediated autoimmune disease, neuromyelitis optica spectrum disorder (NMOSD) [1,2,3]. A compromised endothelial barrier enables the migration of autoantibodies into the target tissue, promoting a local, inflammatory response. While current research predominantly focuses on IgG migration through the brain microvasculature, recent evidence also suggests primary disease activity in retinal tissue following AQP4-IgG interaction with its respective epitope [4,5,6,7,8,9,10,11]. To determine potential inter-tissue disparities in endothelial integrity, it is essential to investigate whether the vascular endothelium in both tissues responds similarly to systemic stressors by compromising the barrier. Identifying a comparable vulnerability of the retinal endothelium to such stressors as observed in brain endothelium could establish the retina as an additional target tissue in NMOSD.

Two such endothelial permeability-modulating stressors are complement system activation products and anaphylatoxins C3a and C5a. Complement activation significantly contributes to NMOSD pathophysiology, and systemic C3a and C5a levels are altered in NMOSD patients [12,13,14,15,16]. Dysregulation of systemic anaphylatoxin levels can lead to increased anaphylatoxic stress on the vessel-lining endothelium, and it is known that both C3a and C5a can provoke endothelial activation [17,18,19,20,21]. This activation can ultimately result in increased paracellular or transcellular permeability, thereby facilitating the transmigration of pathological IgG into the target tissue. Increased paracellular permeability promotes IgG migration through the intercellular space by compromising the tight and adherens junction proteins that regulate molecular and ionic flow. In contrast, increased transcellular permeability enhances IgG traffic directly through the endothelial cells [22]. A differential effect of anaphylatoxin stimulation on the brain and retinal endothelial barrier integrity is plausible, given the high inter-tissue heterogeneity between endothelial cells from different tissues [23].

The primary difference between brain and retinal microvasculature regulation lies in the unique factors influencing retinal blood flow. While the brain maintains stable blood flow through autoregulation, endothelial signalling and neurovascular coupling, the inner retina’s regulation is additionally influenced by light conditions and intraocular pressure [24,25]. The retina’s blood flow significantly changes between light and dark conditions due to varying metabolic demands of the photoreceptors [26]. These differences underscore the inner retina’s unique physiological demands and adaptive regulatory strategies compared to the brain.

In this study, we used primary endothelial cells to compare the stability of the endothelial barrier under anaphylatoxin-mediated stress between two potential target tissues in NMOSD: the brain and the retina. Our findings indicate that the retinal endothelial barrier remains stable under C3a stimulation, whereas brain-derived endothelial cells exhibit increased barrier permeability when exposed to high concentrations of C3a. This study is the first to demonstrate that C5a attenuates the C3a-mediated disruptive effect on the endothelial barrier.

## 2. Results

### 2.1. Anaphylatoxin C3a Induces Concentration-Dependent Paracellular Barrier Breakdown in Cerebral Endothelial Cells While Maintaining Integrity in Retinal Endothelial Cells

Both the brain and the retina are immune-privileged tissues. Disruption of the blood-brain barrier or the inner blood-retinal barrier can lead to significant damage and disease development [27]. In NMOSD, a systemic increase in anaphylatoxin levels can result in stress at the endothelial barriers, which may lead to IgG migration across the damaged barrier [14,15,16]. The objective of this study was to investigate the differential effects of varying C3a and C5a concentrations on the endothelial barrier in both cell types, with the aim of determining whether there is a disparity in tissue-specific protection against anaphylatoxic stress.

We employed the xCelligence real-time cell analysis (RTCA) to observe endothelial paracellular barrier disruption in real time, eliminating the need for cell labelling. To induce anaphylatoxic stress, we treated monocultures of primary human brain microvascular endothelial cells (HBMEC) and primary human retinal endothelial cells (HREC) with C3a or C5a at concentrations of 50 nM, 100 nM or 500 nM during the most stable phase of cell cultivation 17 h post-seeding (Figure 1) [28].

RTCA analysis revealed an initial, anaphylatoxin dose-dependent regulation of the paracellular barrier integrity, indicated by the double normalized cell index (dnCI). Although the short-term regulation of dnCI was observed 30 min after treatment initiation in HBMEC, the 500 nM C3a- and C5a-mediated reduction in paracellular resistance did not reach statistical significance when compared to the untreated control (Figure 1A and Appendix A). In HREC, dnCI increased significantly immediately after the addition of 50 nM C3a and 50 nM, while higher anaphylatoxin concentrations decreased the dnCI (Figure 1C and Appendix A).

Two hours after treatment initiation, 500 nM C5a induced a significant, short-term regulation of paracellular resistance in HBMEC, with a decrease in dnCI of 0.016 ± 0.008 (*p* < 0.05) compared to the untreated control (Figure 1A,B). In contrast, anaphylatoxin stimulation did not alter HREC impedance 2 h after treatment addition compared to the untreated control (Figure 1C,D).

Long-term stimulation with C3a and C5a induced opposing alterations in paracellular resistance in HBMEC and HREC 24 h after treatment initiation. Administration of 500 nM C3a resulted in a decrease in dnCI of 0.032 ± 0.028 (*p* < 0.05) in HBMEC compared to the untreated control (Figure 1A,B). In contrast, 24 h of treatment with C3a increased the dnCI of HREC by 0.071 ± 0.042 at a concentration of 50 nM (*p* < 0.05) and by 0.085 ± 0.031 at a concentration of 100 nM (*p* < 0.01). The increase in endothelial resistance was even more pronounced after C5a stimulation in HREC, where 50 nM C5a increased the dnCI by 0.102 ± 0.053 (*p* < 0.01), and 100 nM C5a increased the dnCI by 0.12 ± 0.045 (*p* < 0.0001), compared to the untreated control (Figure 1C,D).

At 48 h, we observed a similar pattern of anaphylatoxin concentration-dependent endothelial barrier regulation as at the 24 h time point, but with a higher standard deviation (Appendix A).

We observed a tissue-specific effect of 500 nM C3a on the paracellular barrier properties of cerebral and retinal endothelial cells (Figure 1) and consequently wanted to investigate whether this effect extends to the adherens junction marker VE-cadherin (*CDH5*). We examined the gene expression of *CDH5* and the protein expression pattern of VE-cadherin in both cell types at 2 and 24 h after stimulation with 500 nM C3a or 500 nM C5a, respectively (Figure 2).

Quantitative real-time polymerase chain reaction (qRT-PCR) analysis revealed that C3a or C5a treatment did not alter *CDH5* gene expression in HBMEC and HREC in a short- or long-term manner (Figure 2A,B).

Despite the absence of changes in gene expression levels, the protein expression pattern of VE-cadherin demonstrated a correlation with the differential barrier integrity regulation that we had previously identified in the RTCA (Figure 1 and Figure 2C,D).

Two hours after treatment, the endothelial barrier was heterogeneous in untreated and 500 nM C3a-treated HBMEC, with 500 nM C5a-treated cells already showing a more homogeneous VE-cadherin distribution. In HREC, the endothelial barrier was tight and homogeneous after exposure to C3a or C5a. Treatment with 500 nM C5a significantly increased VE-cadherin protein signal compared to the untreated control and to cells treated with 500 nM C3a (*p* < 0.001) (Figure 2C).

Twenty-four hours following treatment, untreated brain and retinal endothelial cells exhibited a continuous monolayer. While VE-cadherin signal was significantly increased in 500 nM C3a-treated HREC compared to the untreated control (*p* < 0.05), this was not the case for HBMEC, where C3a induced a significant reduction in VE-cadherin signal intensity (*p* < 0.01) and a disruption of the endothelial monolayer. In HBMEC, the VE-cadherin signal integrity significantly increased (*p* < 0.001) after 24 h following the addition of 500 nM C5a, and the endothelial monolayer remained unaltered in HREC (Figure 2D).

### 2.2. Anaphylatoxin Treatment Increases C3 Transcript Expression Exclusively in HREC, While C3AR1 mRNA Shows Early, Transient Upregulation in HBMEC Only

C3 is the central complement component, and activation of the complement system leads to proteolytic cleavage into its activation products, C3a and C3b [29]. Increased gene expression of *C3* could result in an increased cleavage and a higher availability of C3a, thereby strengthening the barrier-disruptive effect that C3a exerts on brain endothelial cells (Figure 1A,B). To ascertain the potential regulation of *C3* expression and cleavage, we investigated the gene expression of *C3* and its corresponding receptor, *C3AR1*, in HBMEC and HREC following anaphylatoxin stimulation (Figure 3). Furthermore, we evaluated complement activation by screening for increased C3a presence using immunofluorescent staining (Figure 4).

We observed an opposing regulation of *C3* gene expression in both endothelial cell types at 2 and 24 h post-treatment (Figure 3A,B). In HBMEC, we detected no significant C3a or C5a-mediated regulation of *C3* gene expression at any time point (Figure 3A,B). Conversely, 2 h of C5a exposure induced an increase in *C3* gene expression in HREC when compared with the untreated control (*p* < 0.05) (Figure 3A). Twenty-four hours after treatment initiation, this effect remained stable (*p* < 0.05) and, additionally, we observed a C3a-mediated upregulation in *C3* gene expression compared to the untreated control (*p* < 0.01) (Figure 3B).

The *C3AR1* gene expression pattern after 2 h of treatment showed a similar regulation in HBMEC and HREC after C5a treatment; however, this effect only reached significance in HBMEC (Figure 3C). In detail, 2 h of exposure to 500 nM C5a induced a significant short-term upregulation of *C3AR1* gene expression in HBMEC compared to the untreated control (*p* < 0.01) (Figure 3C). In HREC, 500 nM C5a showed a trend towards upregulation of *C3AR1* gene expression 2 h after treatment initiation. After 24 h, C3a- or C5a-treated HBMEC and HREC did not differ from the untreated control in terms of *C3AR1* gene expression (Figure 3D).

Immunofluorescent staining of C3 revealed an increase in C3 protein signal in both C3a- and C5a-treated HBMEC after 2 h (*p* < 0.01) and in HREC after 2 and 24 h (*p* < 0.05) (Figure 4). C3a significantly decreased C3 signal intensity in HBMEC compared to the untreated control after 24 h (*p* < 0.0019) (Figure 4B).

The pattern of C3 protein expression in HBMEC and HREC after 2 h of anaphylatoxin treatment was consistent with mRNA levels for both C3a and C5a treatments (Figure 3A and Figure 4A). In HREC, the increase in C3 protein levels after 24 h also corresponded with transcript levels. However, at 24 h, discrepancies in protein and mRNA became evident in HBMEC (Figure 3B and Figure 4B). The initial rise in C3 protein in HBMEC after 2 h of anaphylatoxin treatment was followed by a decrease in C3 detection at 24 h, compared to the untreated control, which was not mirrored at the transcript level. This discrepancy in HBMEC due to decreased cellular C3 levels occurred independently of transcription, potentially is a result of increased C3 secretion or enhanced degradation, leading to partial mismatches between protein and mRNA expression (Figure 3A,B and Figure 4.

We detected the complement activation marker C3a in both HBMEC and HREC in both untreated and anaphylatoxin-stimulated conditions. However, the distribution of C3a varied between groups (Figure 4). In HBMEC, 2 h post-treatment, we noticed a broad distribution of C3a, ranging from a spot-wise pattern to covering broader areas in untreated cells. In cells treated with 500 nM C3a and 500 nM C5a, the signal was brighter and more concentrated at a single point. Twenty-four hours after treatment, the distribution of C3a was broader and covered larger areas in both 500 nM C3a and 500 nM C5a-treated cells compared to the 2 h time point and to the untreated control. The C3a protein signal intensity remained stable regardless of treatment and time point in HBMEC (Figure 4).

In HREC, we observed a spot-wise distribution of C3a that was stable across all treatment groups 2 h after treatment. While untreated HREC exhibited a mainly spot-wise distribution of C3a after 24 h, cells treated with 500 nM C3a and 500 nM C5a displayed a broader distribution of C3a signal throughout the cell and showed a significantly increased C3a signal compared to the untreated control (untreated vs. 500 nM C3a: *p* < 0.05; untreated vs. 500 nM C5a: *p* < 0.01), indicating a long-term increasing effect of C3a and C5a treatment on C3a detection in HREC culture (Figure 4). This was in line with *C3* gene expression after anaphylatoxin treatment after 24 h in HREC (Figure 3A,B).

Attempts to stain C3aR in both endothelial cell cultures were unsuccessful.

### 2.3. C3a and C5a Elicit Upregulation of C5 Gene Expression Specifically in HREC, without Affecting Expression Levels in HBMEC

The proteolytic cleavage of the complement component C5 initiates the terminal complement pathway. While the cleavage product C5b will subsequently form the membrane attack complex [30], which will ultimately lead to cell lysis at the site of activation, we investigated whether the second cleavage product and complement activation marker C5a might exert an effect on endothelial integrity.

Using qRT-PCR and immunocytochemistry, we determined a potential regulation of *C5* expression on both the gene and protein level following stimulation with anaphylatoxins C3a and C5a (Figure 5).

We observed a differential regulation of *C5* gene expression in HBMEC and HREC (Figure 5A,B). In HBMEC, *C5* gene expression remained unchanged regardless of anaphylatoxin treatment (Figure 5A,B). In HREC, 500 nM C3a (*p* < 0.05) and 500 nM C5a (*p* < 0.01) significantly upregulated *C5* gene expression in a short-term manner 2 h after treatment initiation (Figure 5A).

Screening for a potential complementary effect on the protein level, we observed an according response of both endothelial cell types to anaphylatoxins (Figure 5C,D). In HBMEC, we localized the C5 protein signal surrounding the cell nucleus and throughout the cell. The C5 signal remained unchanged between treatment groups 2 and 24 h after treatment initiation (Figure 5C,D). In HREC, 2 h after treatment initiation, we observed a distinct C5 distribution pattern which was concentrated on various spots within the endothelial cell in untreated and 500 nM C3a-treated cells. Interestingly, 500 nM C5a treatment induced a cell-spanning, significantly stronger C5 distribution pattern in HREC (untreated vs. 500 nM C5a: *p* < 0.05; 500 nM C3a vs. 500 nM C5a: *p* < 0.01). Twenty-four hours after treatment initiation, C5 signal intensity returned to baseline levels in all treatment groups (Figure 5C,D).

Due to its rapid protein-receptor interaction, C5a has a short half-life and was therefore not analyzed in this context [31].

### 2.4. C5a Treatment Causes an Early Decrease in C5AR1 Protein Detection in HBMEC and a Late Increase in HREC

C5a rapidly interacts with its respective receptor, C5aR1. We investigated the influence of elevated levels of C3a and C5a on the expression of C5aR1 at the gene and protein level using qRT-PCR and immunocytochemistry (Figure 6).

Treatment with 500 nM C3a or 500 nM C5a did not change *C5AR1* expression in HBMEC or HREC (Figure 6A,B).

Interestingly, the C5aR1 protein signal did not correspond with its stable gene expression. After 2 h, we observed a stronger C5aR1 signal in untreated HBMEC than in cells treated with 500 nM C3a or 500 nM C5a (*p* < 0.05). After 24 h, we detected no difference in C5aR1 protein signal strength between treatment groups in HBMEC; however, the distribution pattern shifted from an evenly distributed signal throughout the cell to a spot-wise staining surrounding the cell nucleus (Figure 6C,D). In HREC, the C5aR1 signal exhibited an equal level of intensity across all treatment groups 2 h post-treatment. After 24 h, C5aR1 signal intensity significantly increased in endothelial cells treated with 500 nM C5a compared with the untreated HREC (*p* < 0.05) (Figure 6C,D).

### 2.5. C5a Mitigates the Disruptive Effect of C3a, Demonstrating a Regulatory Role in Maintaining Endothelial Barrier Integrity

Given C3a’s disruptive effect on the brain endothelial barrier and the barrier-enhancing effect of C5a on both endothelial cell types (Figure 1 and Figure 2), we sought to investigate how both anaphylatoxins would affect the barrier in a combinational treatment. Consequently, we conducted a titration of C5a at concentrations of 50 nM, 100 nM and 500 nM against a consistent concentration of 500 nM C3a. We determined the concentration-dependent effect on the endothelial barrier integrity using RTCA and immunocytochemical visualization of VE-cadherin (Figure 7).

Exposure to a combination of C3a and C5a resulted in an immediate regulation of the dnCI (Figure 7A,C and Appendix A). In HREC, treatment with 500 nM C3a + 100 nM C5a significantly increased the dnCI immediately after the addition of treatment compared to 500 nM C3a alone (Appendix A).

Two hours of simultaneous application of C3a and C5a did not affect the paracellular resistance in HBMEC when compared with 500 nM C3a (Figure 7A,B). In HREC, treatment with 500 nM C3a + 100 nM C5a significantly increased the dnCI by 0.01 ± 0.004 (*p* < 0.05) compared to 500 nM C3a alone (Figure 7C,D).

We observed increased dnCI values in almost all treatment groups in HBMEC and HREC after 24 h of combined anaphylatoxin treatment when compared to cells treated with 500 nM C3a alone. A 10:1 ratio of 500 nM C3a + 50 nM C5a significantly attenuated the disruptive effect of 500 nM C3a in HBMEC with an increase in dnCI by 0.146 ± 0.08 (*p* < 0.0001) (Figure 7A,B). Although 500 nM C3a initially did not significantly decrease the dnCI in HREC, we observed an enhancement in paracellular resistance in cells treated with 500 nM C3a + 100 nM C5a by 0.095 ± 0.062 (*p* < 0.05) 24 h after treatment addition when we compared the dnCI to that of the 500 nM C3a-treated HREC (Figure 7C,D).

Forty-eight hours after treatment, the mean dose-dependent regulation of the HBMEC was similar compared with 24 h, but with a higher standard deviation (Appendix A).

Our results indicate a tissue-specific response in endothelial barrier properties to C3a exposure (Figure 1). After determining that this effect can be attenuated by supplementation with 50 nM C5a in HBMEC in the RTCA, we investigated if this effect is reflected in VE-cadherin expression and distribution, using immunocytochemistry.

Two hours after the addition of treatment, we observed a continuous and homogeneous expression of VE-cadherin in all treatment groups (Figure 7E). After 24 h, cells treated with 500 nM C3a + 50 nM C5a continued to display a tight endothelial monolayer with no disruption of the continuous VE-cadherin structure, in contrast to cells treated with 500 nM C3a alone, which exhibited major disruptions of the continuous monolayer. Moreover, 500 nM C3a significantly decreased the VE-cadherin fluorescent signal compared to the untreated control (*p* < 0.01), whereas HBMEC treated with 500 nM C3a + 50 nM C5a maintained a VE-cadherin signal similar to that of the untreated control (Figure 7F).

### 2.6. C3a and C5a Induce Transcellular Permeability in HBMEC but Not in HREC

The primary disease-mediating mechanism in NMOSD is the binding of AQP4-IgG to its respective epitope after migration through the damaged endothelial barrier. To ascertain whether a dysregulation of systemic anaphylatoxin levels might result in the opening of the endothelial barrier for pathological IgG transmigration, we subjected both types of endothelial cells to 500 nM C3a or 500 nM C5a treatment for a duration of 24 h, after which we added purified IgG for 2 h. We investigated the migration of IgG through the endothelial barrier by subjecting abluminal supernatants to Western blot detection (Figure 8A).

In HBMEC, we observed an increased endothelial transcellular permeability in cells treated with both 500 nM C3a and 500 nM C5a, in comparison to the untreated control. C3a treatment resulted in the greatest migration of IgG through the endothelial barrier (*p* < 0.001), but C5a also induced an increased transcellular permeability in HBMEC (*p* < 0.01) (Figure 8B).

In HREC, we detected no significant increase in IgG levels in the abluminal supernatant of any of the treatment groups (Figure 8B).

The results were largely consistent with our previous data, indicating that C3a enhances disruption of the brain but not the retinal endothelial barrier. However, treatment with C5a led to increased transcellular permeability, a change that was not observed in the paracellular RTCA measurements (Figure 1).

## 3. Discussion

There is conflicting evidence regarding the targeting of retinal epitopes in AQP4-IgG seropositive NMOSD [4,5,6,7,8,9,10]. Breakdown of the inner blood–retina barrier may enable a similar translocation of AQP4-IgG into the immune-privileged eye as has been shown in the brain [2,3,32]. The inflammatory drivers and complement activation by-products C3a and C5a are known to be systemically dysregulated in NMOSD patients and have previously been reported to compromise the integrity of the vascular endothelium [12,13,14,15,16]. We investigated the role of C3a and C5a in blood–tissue barrier breakdown to determine whether immune-privileged tissues might possess a differential, tissue-specific resistance to IgG infiltration following exposure to anaphylatoxin-mediated stress.

We report a C3a-mediated decrease in paracellular impedance in HBMEC, whereas low levels of C3a increased transendothelial resistance in HREC. Low levels of C5a stimulation exerted a barrier-enhancing effect in both endothelial cell types. In line with our findings in HBMEC, it has previously been demonstrated that exposure to high concentrations of C3a results in endothelial activation, cell migration, formation of actin stress fibers and in the disruption of the endothelial barrier in HBMEC, human lung microvascular endothelial cells, human umbilical vein endothelial cells (HUVEC) and human dermal microvascular endothelial cells [17,18,19,20,33]. To the best of our knowledge, we are the first ones to report the effect of C3a and C5a on primary human retinal endothelial cells of the inner blood–retina barrier. However, C5a exposure has been investigated in other species and endothelial cell types before, where it exerted no effect on the endothelial barrier integrity or tight junction protein, Zona occludens-1, expression in murine brain microvascular endothelial cells (bEnd3) [34]. In human choroidal endothelial cells, which form the second vital vasculature of the eye in the outer retina, C5a stimulation did not initiate endothelial cell migration or proliferation [35]. However, several studies describe endothelial activation, cell retraction, increased paracellular permeability and apoptosis after C5a stimulation in HUVEC and bEnd3 cells [20,21,36]. A reason for the varying responses to C3a and C5a could be the use of different signalling cascades and cellular localization of C3aR and C5aR in endothelial cell types [20].

Mechanistically, the induction of endothelial hyperpermeability is still under investigation. A likely event involved in endothelial barrier breakdown is the release of calcium from the endoplasmic reticulum and the subsequent increase in intracellular calcium levels, e.g., following C3a-C3aR interaction. Elevated calcium levels can cause a downregulation of VE-cadherin by increasing the expression of endothelial activation marker vascular cell adhesion molecule-1 (VCAM-1) [17]. Besides the upregulation of VCAM-1, the activation of protease-activating receptors (PARs), specifically PAR1, by anaphylatoxins could be another potential barrier-disruptive pathway that has not yet been investigated. Wang et al. previously reported that anaphylatoxin C4a, which is structurally highly similar to C3a and C5a, can bind to PAR1, leading to enhanced endothelial permeability [37]. PAR1 activation leads to the activation of numerous intracellular signalling pathways, ultimately resulting in nitric oxide production, which modulates phosphorylation and thereby downstream internalization of VE-cadherin [38].

The stability of HREC against C3a indicates a tissue-specific protection from complement anaphylatoxin-mediated stress, and thereby potentially from paracellular autoantibody transmigration through the inner blood–retina barrier. A distinct response of the endothelial vasculature from different tissues to stressors can be expected, as organ-specific characteristics in endothelial cells are induced and maintained by the surrounding microenvironment, resulting in a high inter-tissue heterogeneity [23]. Even within one tissue, contour-based 3D image visualization and quantification revealed a high heterogeneity in tight junction protein Claudin-5 protein expression within the murine central nervous system microvasculature [39]. Between different tissues, this becomes specifically evident as the gene transcriptome differs in brain and retinal microvessels in rats [40]. A distinct response to different forms of systemic stress is therefore unsurprising. For example, bovine retinal endothelial cells are more susceptible to glucose-induced oxidative stress than brain-derived endothelial cells, whereas we recently reported a higher susceptibility to oxygen-induced oxidative stress in brain-derived endothelial cells compared to retinal endothelial cells [28,41]. Similarly, alternative complement pathway protein expression, activation and regulation vary highly between human glomerular and brain-derived microvascular endothelial cells, highlighting once more the high inter-tissue heterogeneity in endothelial cells [42]. It remains difficult to discuss the specific influence of C3a and C5a on the endothelium of different tissues, due to the lack of comparative studies in this field, and therefore to make an absolute comparison of abundance and availability of anaphylatoxin receptors in specific endothelial cell types.

Despite the lack of quantitative data about specific protein expression in the endothelium of different tissues, research in the endothelial field has progressed, and endothelial cells have been identified as an extrahepatic source of complement proteins [43]. After exposure to C3a and C5a, we observed an upregulation in C3 and C5 transcript and protein detection in HREC, but only for C3 protein in HBMEC. Taken together with the endothelial barrier stability in HREC under anaphylatoxic stress, this points towards a complement-mediated protection of the microvasculature of the inner retina. It has previously been reported that HBMEC and HUVEC express *C3* and *C5* at the transcript level [42,43,44,45], whereas HREC were only found to express *C3* [46]. Research regarding the influence of complement anaphylatoxins on the regulation of these complement genes is limited, though it has been reported that C3a exposure increased *C3* expression in normal human epidermal keratinocytes [47]. While we determined a regulation of C3 and C5 expression in HREC, we observed a significant increase in *C3AR1* expression following 2 h of C5a stimulation in HBMEC but not HREC. *C5AR1* gene expression was stable in both cell types regardless of treatment or duration. Endothelial anaphylatoxin receptor expression has been described before in HUVEC, HBMEC, mouse dermal microvascular endothelial cells and choroidal endothelial cells [19,20,35,48]. Specifically in HUVEC, exposure to 100 nM C3a and 100 nM C5a resulted in an upregulation of *C5AR1* and, similar to our findings in HBMEC, *C3AR1* gene expression [19]. Additionally, in mesenchymal stem cells, macrophages and tumor cells, C3a increases the expression of its own receptor in an autocrine manner [49]. This indicates a form of cross-talk between the C5a-C5aR and C3a-C3aR pathways. In fact, we report that after 2 h of C3a and C5a treatment, a reduction in C5aR1 protein detection is observed in HBMEC. This C5aR1 downregulation is accompanied by a potential compensatory upregulation of *C3aR* at the transcript level in HBMEC. While this upregulation shows only a trend for C3a, it is statistically significant for C5a, indicating a transient increase in *C3aR* transcription. This may initiate an amplification of the C3a-mediated effect within brain endothelial cells. In contrast, HREC did not exhibit early adjustments in the expression of anaphylatoxin receptors, either at the protein or transcript level. As a possible consequence, the intact HREC barrier may provide initial insights into the differential mechanisms underlying endothelial cell type-specific susceptibility to systemic complement-related stress.

Besides the ability of C3a and C5a to differentially regulate gene expression in endothelial cells, one main finding of our study is that a combinational treatment of 500 nM C3a and 50 nM C5a prevents the barrier-disruptive effect mediated by C3a treatment alone in HBMEC. While elevated C5a levels reduced the disruptive effect of C3a on HBMEC, only the lowest concentration of C5a significantly increased paracellular impedance. This suggests that small amounts of C5a help maintain the integrity of endothelial cells, preventing them from becoming ‘leaky’. However, when C5a levels become too high, the barrier becomes compromised again. This highlights the importance of not completely blocking C5a, but rather maintaining it at low levels, as achieved with the monoclonal C5 inhibitor Eculizumab in NMOSD [50]. A balance must be maintained, much like the natural regulation of the complement cascade, where C3a is produced in significantly higher amounts early in the cascade, while C5a appears later in much smaller quantities.

The literature is sparse about the overcompensating effect of C5a on C3a. Both C3a and C5a exhibit synergistic interactions due to their roles in inflammatory processes. However, recent studies on kidney diseases have demonstrated that C3a and C5a can also function antagonistically. Specifically, C3a primarily displays anti-inflammatory effects, whereas C5a serves as a pro-inflammatory mediator [51,52].

Since the intracellular calcium release from the endoplasmic reticulum and the downstream downregulation of adherens junction protein VE-cadherin seem to be major contributors to endothelial hyperpermeability after C3a–C3aR interaction, a potential reason for a rescue effect mediated by C5a could be an altered cytosolic calcium release after co-administration of C3a and C5a, as has already been described in retinal pigment epithelial cells [53]. This indicates a differential effect of both anaphylatoxins in regulating intracellular calcium levels and associated paracellular barrier integrity.

Hence, we mimicked luminal IgG presence in vitro to determine whether a dysregulation in anaphylatoxin presence could influence IgG migration through the endothelial barrier into the abluminal space.

We found that IgG transmigrates across the brain endothelial barrier after exposure to high levels of C3a and C5a, whereas we did not detect IgG transmigration through the retinal endothelial barrier after anaphylatoxin treatment. The absence of IgG in abluminal supernatants of HREC was largely consistent with our findings in the RTCA, suggesting no effect of anaphylatoxins on paracellular and transcellular pathological IgG transmigration into the retina. In HBMEC, our data suggest that C5a decreases the transcellular barrier integrity, as determined by diffusion assays. However, C5a had a less-pronounced disruptive influence on paracellular resistance in brain endothelial cells, as observed in RTCA. An increase in transcellular permeability is therefore the likely consequence following endothelial activation after C5a exposure in HBMEC [20,21,36]. Activation of endothelial cells is known to upregulate various receptors, including the Fcγ receptor [54]. This receptor binds the Fc part of IgG and can transport IgG through the endothelial barrier [55]. Increased C5a levels in NMOSD patients could thereby enhance the transcytosis of pathological IgG into the brain.

The explorative nature of this study is certainly a limitation, considering the use of an endothelial monoculture, without the tissue-specific microenvironment mediated by other cell types like pericytes, astrocytes or Müller cells. The absence of in vivo-like shear stress, which induces endothelial-specific properties, like the expression of complement surface regulators and inhibitors CD46, CD55 and CD59, is another limitation [56]. Due to the individual genotype of the cells, analysis of more endothelial cells from various donors would strengthen these initial findings. Additionally, although C5a enhanced paracellular endothelial impedance, it is possible that C5a may concurrently promote transcellular IgG transport in HBMEC. Although we observed a more stable and tighter barrier following the combined treatment of 500 nM C3a and 50 nM C5a, we are unable to draw conclusions regarding the transendothelial transport of pathological IgG after co-administration of C3a and C5a. This observation introduces a potential new area of investigation into the role of anaphylatoxins in regulating transendothelial IgG transport.

We conclude that C3a impairs the endothelial barrier in human brain microvascular endothelial cells but not in endothelial cells of the inner retinal microvasculature. This indicates that the brain might be more susceptible to pathological IgG translocation through a damaged endothelial barrier than the retinal tissue in NMOSD patients with dysregulated systemic C3a levels. This disruptive effect on cellular level can be prevented by addition of C5a, indicating an overcompensating effect of C5a on C3a and that the availability of anaphylatoxins at a right concentration is vital for a functional endothelial barrier. Gene regulation of *C3* and *C5* remains stable after C3a and C5a exposure in HBMEC but increases in HREC, indicating a protective role of complement in barrier maintenance. Upregulation of *C3AR1* in HBMEC but not HREC after C5a stimulation points towards a connection between C5aR activation and *C3AR1* expression. As a future outlook, it would be sensible to include transcriptomic and proteomic analyses to gain a deeper understanding of the cell-specific mechanisms underlying the response to anaphylatoxic stress. Gene transcription analysis of complement inhibitory surface proteins could be a future approach to assess differential expression in both endothelial cell types, which might determine whether there is a tissue-specific protection from complement-mediated bystander injury. Furthermore, replicating these treatments using a 3D cell culture system that incorporates barrier-specific cell types, such as pericytes, astrocytes or Müller cells, will enhance endothelial characteristics and reinforce the findings of this exploratory study.

## 4. Materials and Methods

### 4.1. Cell Culture and Treatment

HBMEC and HREC were cultured under uniform conditions. HBMEC, sourced from Cell Systems (Kirkland, WA, USA, #ACBRI 376), and HREC, obtained from Innoprot (Derio, Bizkaia, Spain, #P10880), were thawed and seeded into T75 flasks coated with either attachment factor (for HBMEC, 1 mL/10 cm^2^, Cell Systems, Kirkland, WA, USA, #4Z0-210) or fibronectin (for HREC, 2 µg/cm^2^, Innoprot, Derio, Bizkaia, Spain, #P8248) for expansion. Both cell types were maintained in their respective media (for HBMEC, EGM2-MV, Lonza Group, Basel, Switzerland, #CC-3156, for HREC, endothelial cell medium, Innoprot, Derio, Bizkaia, Spain, #P60104) until reaching 85% confluency. Subsequently, HBMEC and HREC underwent two passages (1:6 for HBMEC, 1:5 for HREC). For splitting, both cell types were trypsinized using a trypsin–EDTA solution (KGaA, Darmstadt, Germany, #T4174) for 8 min at 37 °C for HBMEC and 2 min at 37 °C for HREC, followed by centrifugation (600× *g* for 5 min for HBMEC, 193 × *g* for 5 min for HREC). Both cell types were expanded to 85% confluency before freezing passages 5–6 (50% EGM2-MV, 40% FCS, 10% DMSO). Passages 6–7 were utilized for all experiments involving HBMEC and HREC. The cells were cultivated at 37 °C in a humidified atmosphere with 5% CO_2_.

For all experiments, cells were thawed into a T175 flask and grown to reach 85% confluency before seeding HBMEC and HREC onto 12-well filter inserts (0.4 µm pore, Greiner Bio-One GmbH, Frickenhausen, Germany, #653393) at densities of 1 × 10^5^ cells/cm^2^ for HBMEC and 1.5 × 10^5^ cells/cm^2^ for HREC. The inserts were coated with either attachment factor for HBMEC or fibronectin for HREC as previously described [28]. Following a 17 h growth phase, HBMEC and HREC were subjected to recombinant C3a (R&D Systems, Minneapolis, MN, USA, #3677-C3-025) or recombinant C5a (R&D Systems, Minneapolis, MN, USA, #2037-C5-025/CF) treatment for 2 and 24 h. Both anaphylatoxins were diluted in the respective culture medium. Treatment was administered at concentrations of 50 nM, 100 nM and 500 nM for RTCA and 500 nM for immunocytochemical staining and qRT-PCR samples. Unaltered culture medium was added to the untreated control. Following treatment, the cells underwent detailed analysis through immunocytochemistry, RTCA, and Western blot, as described in the subsequent sections.

### 4.2. RTCA Using the xCelligence Impedance Sensing System

The xCelligence real-time cell analysis system (OMNI Life Science GmbH & Co KG, Bremen, Germany) was employed to conduct real-time monitoring of cell growth, adhesion and paracellular barrier function using an impedance sensing system. HBMEC and HREC underwent trypsinization and were seeded on attachment factor-coated E-16 plates (for HBMEC, 1 mL/10 cm^2^, Cell Systems, Kirkland, WA, USA, #4Z0-210) or fibronectin-coated E-16 plates (for HREC, 2 µg/cm^2^, Innoprot, Derio, Bizkaia, Spain, #P8248), with cell densities of 1 × 10^5^ cells/cm^2^ for HBMEC and 1.5 × 10^5^ cells/cm^2^ for HREC [28]. Impedance changes, represented as CI, were calculated by subtracting the baseline impedance from the impedance at a specific time point and dividing it by the nominal impedance value. The alterations in CI were continuously monitored over 72 h with minute-by-minute measurements.

C3a and C5a treatment was introduced after allowing cells to settle for 17 h, and monitoring was continued for additional 55 h. All molecules used for treatment were diluted in the respective medium, with unaltered culture medium serving as a control. The CI of each curve was double-normalized to its respective CI value measured before the addition of anaphylatoxin treatment and to the untreated control.

### 4.3. Quantitative Real-Time Polymerase Chain Reaction

HBMEC and HREC were harvested at 2 h and 24 h post-treatment. mRNA extraction was performed using the NucleoSpin RNA kit (Macherey-Nagel, Düren, Germany, #740955). Following elution, mRNA was reverse transcribed to cDNA using the QuantiTect Reverse Transcription kit (Qiagen, Hilden, Germany, #205311). For gene expression analysis, qRT-PCR was conducted using the Power SYBR green PCR Master Mix (Thermo Fisher Scientific, Waltham, MA, USA, #10209284) and primers for the gene-specific quantification of *CDH5*, *C3*, *C3AR1*, *C5* and *C5AR1* (Table 1 and Table 2). All in-house designed primers were validated to specifically detect only their respective targets (Appendix A). The QuantStudioTM 5 Real-Time PCR System (Thermo Fisher Scientific, Waltham, MA, USA) was used for analysis. The amplification protocol consisted of 40 cycles with an annealing temperature of 60 °C for 1 min. Glyceraldehyde 3-phosphate dehydrogenase (*GAPDH*) expression served as the normalization reference for gene expression (Table 1). The corresponding fold change between control and treatment groups was calculated utilizing the 2(−ΔΔCt) method [57]. qPCR data reflect the median of a sample size of *n* = 4–8, as represented in the corresponding graphs.

### 4.4. Immunocytochemistry

HBMEC and HREC were cultured on 12-well filter inserts as previously described. Following 2 and 24 h of anaphylatoxin exposure, both HBMEC and HREC were fixed using 4% paraformaldehyde for 20 min at 4 °C. The samples were then washed three times with phosphate-buffered saline (PBS), followed by permeabilization with PBS containing 1% Tween-20 for 45 min. Subsequently, the cells were blocked with a solution of 3% bovine serum albumin (BSA) and 1% Tween-20 in PBS for 1 h at room temperature. Cells were then incubated in blocking solution containing primary antibodies overnight at 4 °C. The primary antibodies used were as follows: mouse anti-VE-cadherin (2.5 µg/mL, R&D Systems, Minneapolis, MN, USA, #MAB9381), rabbit anti-C3 (1 µg/mL, Abcam, Cambridge, UK, #ab181147), mouse anti-C3a (2 µg/mL, Hycultec GmbH, Beutelsbach, Germany, #HM2074), goat anti-C5 (1:300, Complement Technology, Inc., Tyler, TX, USA, #A220) and rabbit anti-C5aR1 (2.13 µg/mL, Proteintech, Manchester, UK, #21316-1-AP).

After primary antibody staining, cells were washed three times with PBS and then incubated for 45 min with the appropriate secondary antibodies (donkey IgG anti-mouse IgG (H+L)-Alexa Fluor™ 488 (3 µg/mL, Jackson ImmunoResearch Labs, West Grove, PA, USA, #715-545-150), goat anti-rabbit IgG (H+L)-Alexa Fluor™ 546 (4 µg/mL, Thermo Fisher Scientific, Waltham, MA, USA, #A-11010), donkey anti-goat IgG (H+L)-Cy3 (3 µg/mL, Jackson ImmunoResearch Labs, West Grove, PA, USA, #705-165-147) in 3% BSA in PBS. Cells were washed three times with PBS before mounting with fluorescence mounting medium (Agilent, Santa Clara, CA, USA, #S302380-2). The samples were scanned using the confocal microscope Leica SP8 (Leica Microsystems GmbH, Wetzlar, Germany) with a x20 objective lens. Image processing and analysis of the mean FI was carried out using FIJI, version 2.14.0 (29). The mean FI was normalized to the untreated control and is displayed as relative mean FI.

### 4.5. IgG Purification

To obtain purified IgG for endothelial transbarrier assays, 2 mL pooled human complement serum (Innovative Research, Inc., Novi, MI, USA, #ICSER) were subjected to protein G spin column (Thermo Fisher Scientific, Waltham, MA, USA, #89957) purification. Briefly, the column was equilibrated twice by adding 2 mL binding buffer and centrifuging for 1 min at 1000× *g*. The serum was applied to the emptied column and incubated with end-over-end mixing for 10 min. Following a 1-min centrifugation at 1000× *g*, the column was washed three times with 2 mL binding buffer, each followed by centrifuging for 1 min at 1000× *g*. To elute the protein G-bound IgG, 100 µL neutralization buffer was added to a collection tube, followed by the addition of 1 mL elution buffer to the column, which was then placed into the collection tube and centrifuged for 1 min at 1000× *g*. The eluted sample was then dialyzed against 2 L PBS using a dialysis membrane (Carl Roth GmbH & Co. KG, Karlsruhe, Germany, #E683.a) for 1 h at room temperature, followed by another incubation step overnight at 4 °C and a final 1-h incubation at room temperature with a buffer change after each incubation step. The solution volume was then reduced at 4000× *g* to obtain an IgG concentration of 10 mg/mL, using centrifugal filter units (Merck KGaA, Darmstadt, Germany, #UFC8100). The relative absorbance was measured at 280 nm using the NanoPhotometer N60 (Implen GmbH, Munich, Germany).

### 4.6. Transbarrier Assay

HBMEC and HREC were cultivated on 12-well filter inserts as previously described [28]. At 17 h post seeding, cells were treated with 500 nM C3a or 500 nM C5a, diluted in culture medium. An untreated control was supplemented with an equal amount of unconditioned HBMEC or HREC medium. At 24 h after the addition of treatment, all wells were supplemented luminally with 0.1 mg/mL purified IgG, diluted in culture medium. Two hours later, abluminal supernatants were harvested to determine the extent of IgG migration through the endothelial barrier using Western blot detection.

### 4.7. Western Blot

For IgG detection, abluminal supernatants were concentrated to a volume of 200 µL using centrifugal filter units (Merck KGaA, Darmstadt, Germany, #UFC5100). 15 µL of the concentrated supernatant samples were separated under non-reducing conditions, utilizing an 8% gel in SDS-PAGE (Bio-Rad Laboratories Inc., Hercules, CA, USA). Protein transfer onto a PVDF membrane (Merck KGaA, Darmstadt, Germany, #IEVH85R) was performed using a wet blot system (Bio-Rad Laboratories Inc., Hercules, CA, USA), followed by blocking for 1 h with a blocking buffer (5% skim milk powder, 0.1% Tween 20 in Tris-buffered saline). The membrane was incubated overnight at 4 °C with detection antibodies diluted in blocking buffer (goat anti-human IgG (H+L)-Alexa Fluor™ 647, 2 µg/mL, Thermo Fisher Scientific, Waltham, MA, USA, #A-21445). After three washing steps with washing buffer (1% Tween 20 in Tris-buffered saline), the fluorescent signal was detected using the ChemiDoc MP imaging system (Bio-Rad Laboratories, Inc., Hercules, CA, USA). Western blot images were analyzed conducted using FIJI, version 2.14.0 [58].

### 4.8. Statistics

The datasets for 2 and 24 h were analyzed separately, treating anaphylatoxin concentration as an independent variable. Preliminary screening of the datasets was conducted to identify outliers using the robust regression and outlier method. The Gaussian normal distribution of all data was examined using the Shapiro–Wilk test, with a significance level set at α = 0.05. To assess the homogeneity of variance for data assumed to follow a normal distribution, the Brown–Forsythe test was applied. Once equal variance within the dataset was confirmed, a one-way ANOVA was performed. Subsequent multiple comparisons were carried out using Dunnett’s T3 *post hoc* test.

For datasets where the assumption of normal distribution was violated, the Kruskal–Wallis test was employed, and *post hoc* comparisons were conducted using Dunn’s multiple comparisons test. The level of statistical significance was set at *d* < 0.05. All statistical analyses were executed using GraphPad Prism Version 9.3.1 (GraphPad Software, Inc., Boston, MA, USA).

## Figures and Tables

**Figure 1 ijms-25-11240-f001:**
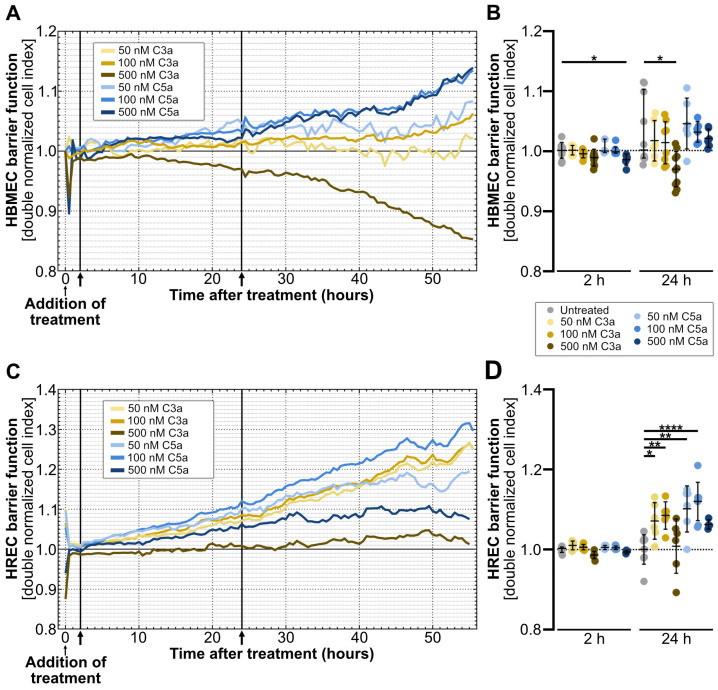
Anaphylatoxins C3a and C5a differentially regulate endothelial monoculture dnCI in a dose-dependent manner. (**A**,**B**) Monitoring of dnCI changes in response to 50 nM, 100 nM and 500 nM of anaphylatoxins C3a and C5a revealed a transient decrease in dnCI by 500 nM C5a after 2 h of treatment in HBMEC. At 24 h after anaphylatoxin stimulation, 500 nM C3a significantly decreased the paracellular resistance in HBMEC compared to the untreated control. (**C**,**D**) In HREC, we observed no significant alterations in paracellular resistance after 2 h of C3a or C5a treatment. A dose-dependent increase in dnCI occurred 24 h after treatment initiation in HREC treated with 50 nM C3a, 100 nM C3a, 50 nM C5a and 100 nM C5a compared to the untreated control. (**A**,**C**) ↑ indicates analyses time points at 2 and 24 h. All data were normalized to the respective cell index (CI) value before the addition of treatment and to the untreated control (dnCI). (**A**,**B**) RTCA graph represents data from *n* = 10 for untreated and 500 nM C3a, *n* = 6 for 50 nM C3a and 50, 100 and 500 nM C5a and *n* = 8 for 100 nM C3a treated cells, respectively. (**C**,**D**) RTCA graph represents data from *n* = 9 for untreated, *n* = 6 for 50 nM C5a and 50, 100 and 500 nM C3a, *n* = 8 for 100 nM C5a and *n* = 4 for 500 nM C5a, respectively. (**B**,**D**) Two- and twenty-four-hour time points were analyzed as separate datasets. Kruskal–Wallis test with Dunn’s multiple comparisons test was used for datasets including non-parametric data. One-way analysis of variance (ANOVA) with Dunnett’s T3 *post hoc* test was used for parametric data sets. * *p* < 0.05, ** *p* < 0.01, **** *p* < 0.0001.

**Figure 2 ijms-25-11240-f002:**
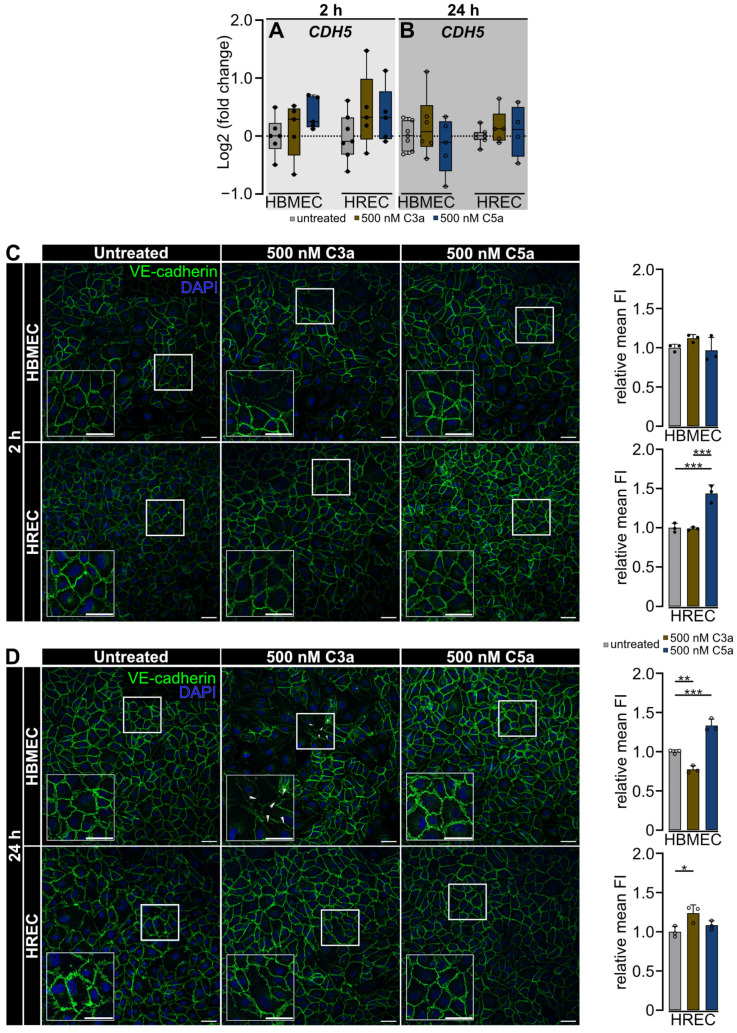
C3a and C5a do not change *CDH5* gene expression but alter VE-cadherin on protein level in HBMEC and HREC. (**A**,**B**) Treatment with 500 nM C3a or 500 nM C5a did not change *CDH5* gene expression in HBMEC and HREC compared with the untreated control. (**C**) Two hours after treatment, HBMEC exhibited a heterogeneous VE-cadherin phenotype in untreated and 500 nM C3a-treated cells. HBMEC treated with 500 nM C5a displayed a more homogeneous distribution of VE-cadherin. VE-cadherin mean fluorescent intensity (mean FI) remained stable regardless of treatment. Two hours after treatment, HREC displayed a homogeneous monolayer in all treatment groups without disruption of the endothelial monolayer. VE-cadherin signal intensity was increased by 500 nM C5a. (**D**) After 24 h, HBMEC displayed a homogeneous phenotype in untreated and 500 nM C5a-treated cells, but they were elongated and showed disrupted VE-cadherin expression in 500 nM C3a-treated cells (white arrowheads). VE-cadherin signal intensity was decreased by 500 nM C3a but was increased by 500 nM C5a. In HREC, the homogenous phenotype remained stable over 24 h. 500 nM C3a increased VE-cadherin signal intensity. (**A**,**B**) Data were curated from 4 separate cultures as technical replicates. Two- and twenty-four-hour time points were analyzed as separate datasets. One-way ANOVA with Dunnett’s T3 *post hoc* test. (**C**,**D**) Three separate cultures as technical replicates. White boxes visualise magnified areas. Scale bar 50 µm. Kruskal-Wallis test for non-parametric data sets, one-way ANOVA with Dunnett’s T3 *post hoc* test for parametric data sets. * *p* < 0.05, ** *p* < 0.01, *** *p* < 0.001.

**Figure 3 ijms-25-11240-f003:**
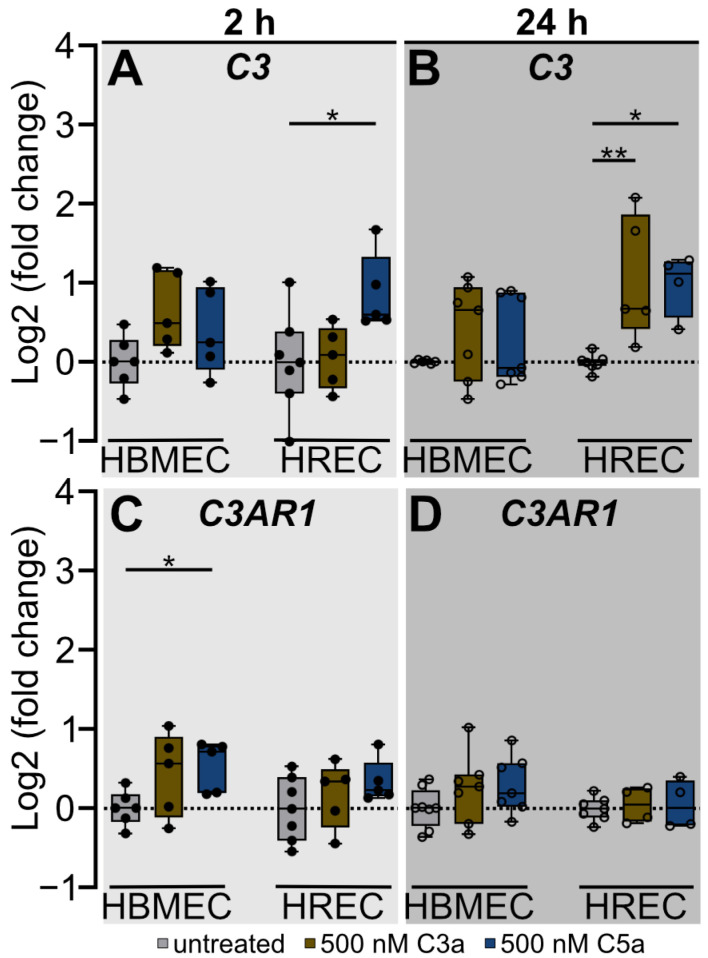
Anaphylatoxin treatment caused increased *C3* expression in HREC and increased *C3AR1* expression in HBMEC. (**A**,**B**) qRT-PCR analysis revealed no effect of anaphylatoxin treatment on *C3* expression in HBMEC but disclosed an increase in *C3* expression in 500 nM C5a-treated HREC 2 h post-treatment and a 500 nM C3a and 500 nM C5a mediated increase in *C3* gene expression 24 h post-treatment in HREC. (**C**,**D**) Two hours’ exposure to 500 nM C5a increased *C3AR1* expression in HBMEC, while anaphylatoxin treatment had no significant effect on *C3AR1* expression in HREC. (**A**–**D**) Data were collected from 4 separate cultures as technical replicates. Two- and twenty-four-hour time points were analyzed as separate datasets. Kruskal–Wallis test with Dunn’s multiple comparisons test for datasets including non-parametric data. One-way ANOVA with Dunnett’s T3 *post hoc* test for parametric datasets. * *p* < 0.05, ** *p* < 0.01.

**Figure 4 ijms-25-11240-f004:**
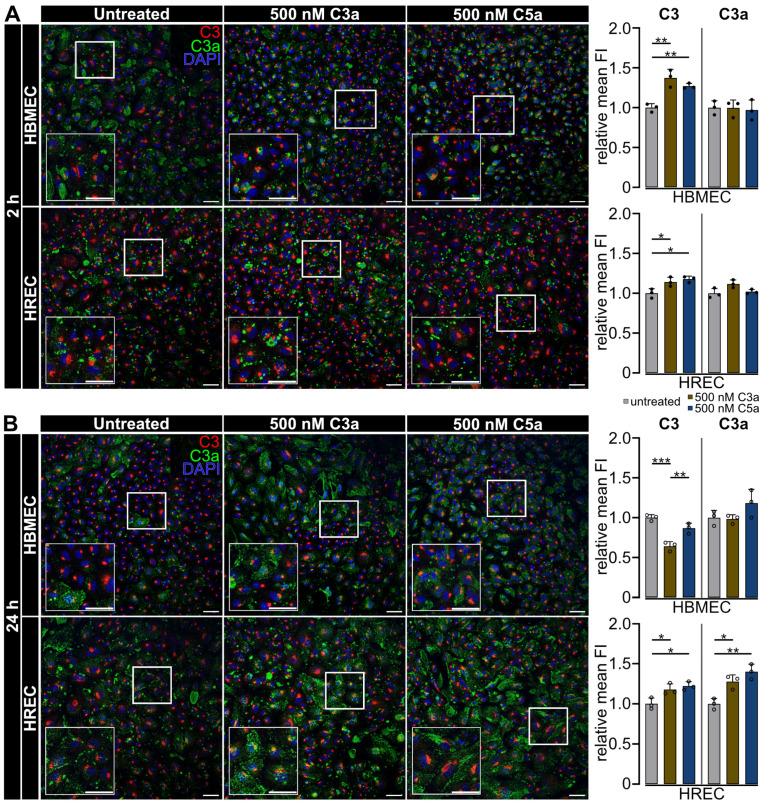
C3a presence increased in HREC after exposure to C3a and C5a. (**A**) After 2 h, the C3 protein signal intensity increased in C3a- and C5a-treated HBMEC. The C3a signal in untreated HBMEC exhibited an uneven distribution pattern, ranging from spot-wise signals to broader distribution. In HREC, C3 protein expression pattern in close proximity to the nucleus remained unchanged across all treatment groups after 2 h, but the signal intensity increased in C3a- and C5a-treated HREC. HBMEC and HREC C3a signal intensity remained stable across treatment groups and exhibited a spot-wise pattern across all treatment groups after 2 h. (**B**) After 24 h, the C3 signal decreased in C3a-treated HBMEC and returned to baseline in C5a-treated cells. In untreated HBMEC, C3a exhibited an uneven distribution pattern, ranging from spot-wise signals to broader distribution 24 h post-treatment initiation. The localization of C3a transitioned from spot-wise distribution to a more diffuse cellular distribution in HBMEC treated with 500 nM C3a and 500 nM C5a 24 h after treatment initiation. HREC C3 protein expression pattern in close proximity to the nucleus remained unchanged across all treatment groups. C3 signal intensity increased in C3a- and C5a-treated HREC after 24 h. The spot-wise C3a distribution pattern seen after 2 h persisted in untreated HREC 24 h after treatment but expanded to a broader, cell-covering signal in cells treated with 500 nM C3a and 500 nMC5a. C3a signal intensity increased in C3a- and C5a-treated HREC 24 h after treatment initiation. (**A**,**B**) Three separate cultures as technical replicates. White boxes visualise magnified areas. Scale bar 50 µm. Kruskal-Wallis test for non-parametric data sets, one-way ANOVA with Dunnett’s T3 *post hoc* test for parametric data sets. * *p* < 0.05, ** *p* < 0.01, *** *p* < 0.001.

**Figure 5 ijms-25-11240-f005:**
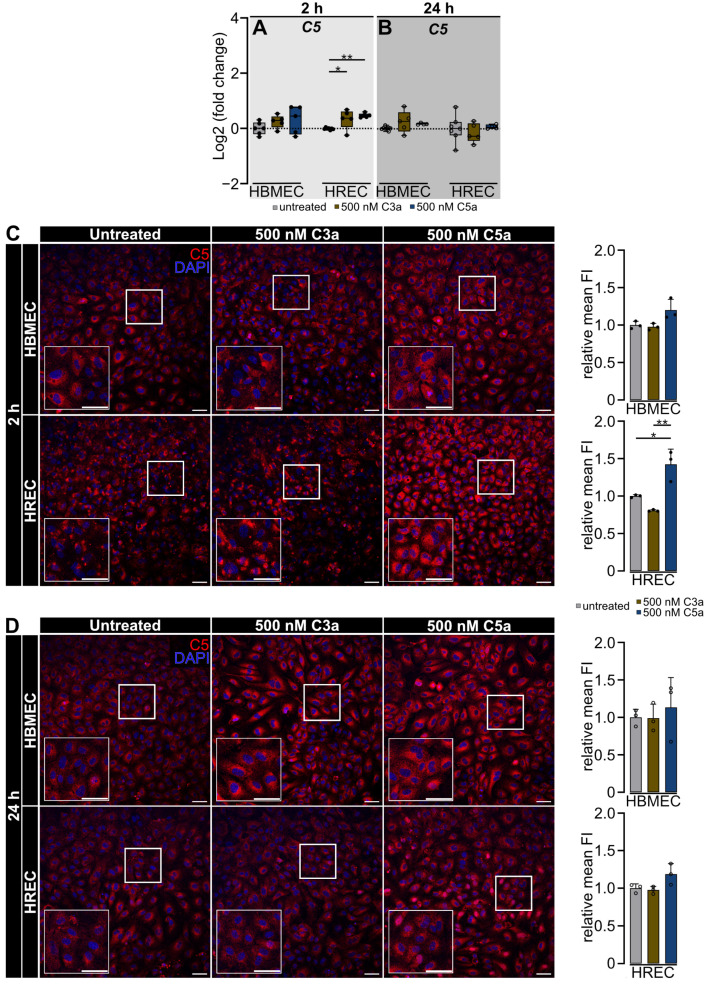
C5 gene and protein expression are elevated in HREC following C5a stimulation, whereas no significant changes are observed in HBMEC. (**A**,**B**) qRT-PCR indicated no variations in *C5* gene expression among treatment groups or analysis time points at 2 and 24 h in HBMEC. Two hours post-treatment, exposure to 500 nM C3a and 500 nM C5a resulted in a significant upregulation in *C5* gene expression in HREC. This effect was reversed 24 h post-treatment. (**C**,**D**) Stimulation with 500 nM C3a and 500 nM C5a provoked no change in C5 protein signal strength in HBMEC. Two hours post-treatment, C5 exhibited a spot-wise expression pattern in both untreated and 500 nM C3a-treated HREC, whereas HREC treated with 500 nM C5a showed a broader distribution of immunofluorescent signals. C5a increased C5 signal intensity compared to the untreated control and C3a-treated HREC. After 24 h, C5 detection signal returned to baseline in C3a- and C5a-treated cells. (**A**,**B**) Data were collected from 4 separate cultures as technical replicates. Two- and twenty-four-hour time points were analyzed as separate datasets. One-way ANOVA with Dunnett’s T3 *post hoc* test. (**C**,**D**) Three separate cultures as technical replicates. White boxes visualise magnified areas. Scale bar 50 µm. One-way ANOVA with Dunnett’s T3 *post hoc* test. * *p* < 0.05, ** *p* < 0.01.

**Figure 6 ijms-25-11240-f006:**
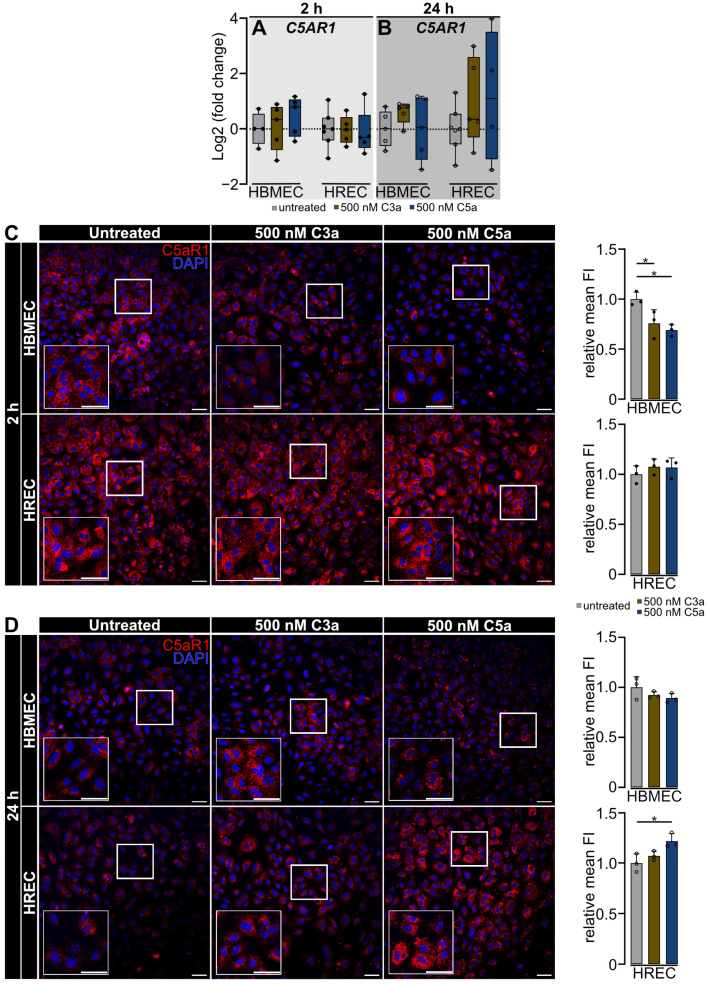
*C5AR1* transcript expression remains unchanged, but protein detection changes under anaphylatoxic stress. (**A**,**B**) qRT-PCR analysis showed no difference in *C5AR1* gene expression between untreated and C3a- or C5a-treated HBMEC and HREC. (**C**) Treatment with 500 nM C3a and 500 nM C5a reduced C5aR1 protein signal in HBMEC after 2 h of treatment. Immunofluorescent staining of C5aR1 in HREC showed a stable and comparable signal after 2 h in all treatment groups. (**D**) After 24 h, the C5aR1 signal was stable between treatment groups in HBMEC. C5aR1 protein signal intensity increased in 500 nM C5a-treated HREC after 24 h. (**A**,**B**) Data were curated from 4 separate cultures as technical replicates. Two- and twenty-four-hour time points were analyzed as separate datasets. One-way ANOVA with Dunnett’s T3 *post hoc* test. (**C**,**D**) Three separate cultures as technical replicates. White boxes visualise magnified areas. Scale bar 50 µm. One-way ANOVA with Dunnett’s T3 *post hoc* test. * *p* < 0.05.

**Figure 7 ijms-25-11240-f007:**
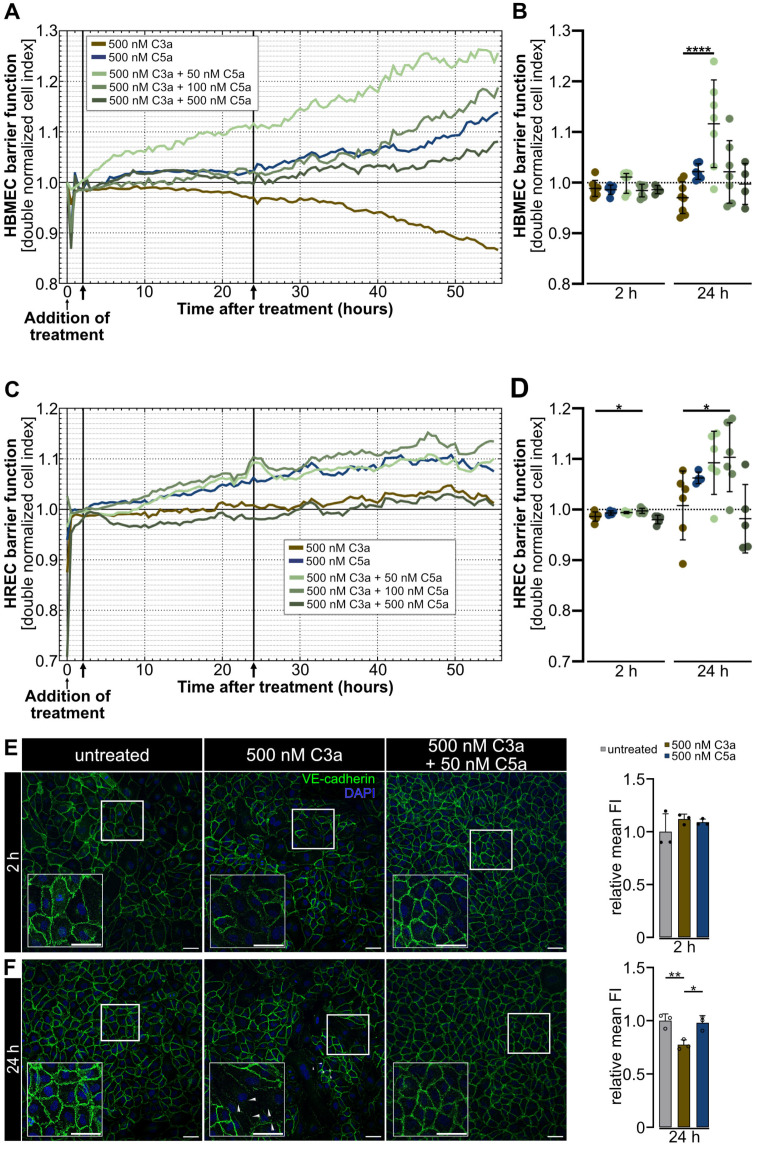
Adding 50 nM of C5a, but not 100 nM or 500 nM, counteracted the barrier-disruptive effect of 500 nM C3a and improved the HBMEC barrier after 24 h. (**A**,**B**) Monitoring of combined anaphylatoxin-treated dnCI changes in HBMEC revealed a C5a dependent increase in dnCI, which counteracted the barrier-disruptive effect of C3a. 24 h post-anaphylatoxin stimulation, HBMEC treated with 500 nM C3a + 50 nM C5a reached a significantly higher dnCI compared to 500 nM C3a-treated cells. (**C**,**D**) RTCA measurement revealed an increase in dnCI in HREC treated with 500 nM C3a + 100 nM C5a 2 h after treatment addition compared to 500 nM C3a-treated cells. This regulation was maintained for 24 h after exposure to treatment. (**E**) Immunocytochemistry revealed a homogeneous endothelial monolayer in all treatment groups in HBMEC, 2 h post-treatment. (**F**) Twenty-four hours post-treatment, 500 nM C3a decreased the VE-cadherin signal intensity, whereas treatment with 500 nM C3a + 50 nM C5a obtained a VE-cadherin signal strength similar to that of the untreated control. (**A**,**C**) ↑ indicate analyses time points 2 and 24 h. All data were normalized to the respective CI value before addition of treatment and to the untreated control. (**A**,**B**) RTCA graph represents the mean of data from *n* = 9 for 500 nM C3a, *n* = 6 for 500 nM C5a and *n* = 7 for 500 nM C3a + 50 nM C5a and 500 nM C3a + 100 nM C5a and *n* = 4 for 500 nM C3a + 500 nM C5a-treated cells, respectively. (**C**,**D**) RTCA graph represents the mean of data from *n* = 6 for 500 nM C3a, 500 nM C3a + 50 nM C5a and 500 nM C3a + 100 nM C5a and *n* = 4 for 500 nM C5a and *n* = 5 for 500 nM C3a + 500 nM C5a. (**B**,**D**) Two- and twenty-four-hour time points were analyzed as separate datasets. Kruskal–Wallis test with Dunn’s multiple comparisons test for datasets including non-parametric data. One-way ANOVA with Dunnett’s T3 *post hoc* test for parametric datasets. (**E**,**F**) Three separate cultures as technical replicates. White boxes visualise magnified areas. Scale bar 50 µm. Kruskal-Wallis test for non-parametric data sets, one-way ANOVA with Dunnett’s T3 *post hoc* test for parametric data sets. * *p* < 0.05, ** *p* < 0.01, **** *p* < 0.0001.

**Figure 8 ijms-25-11240-f008:**
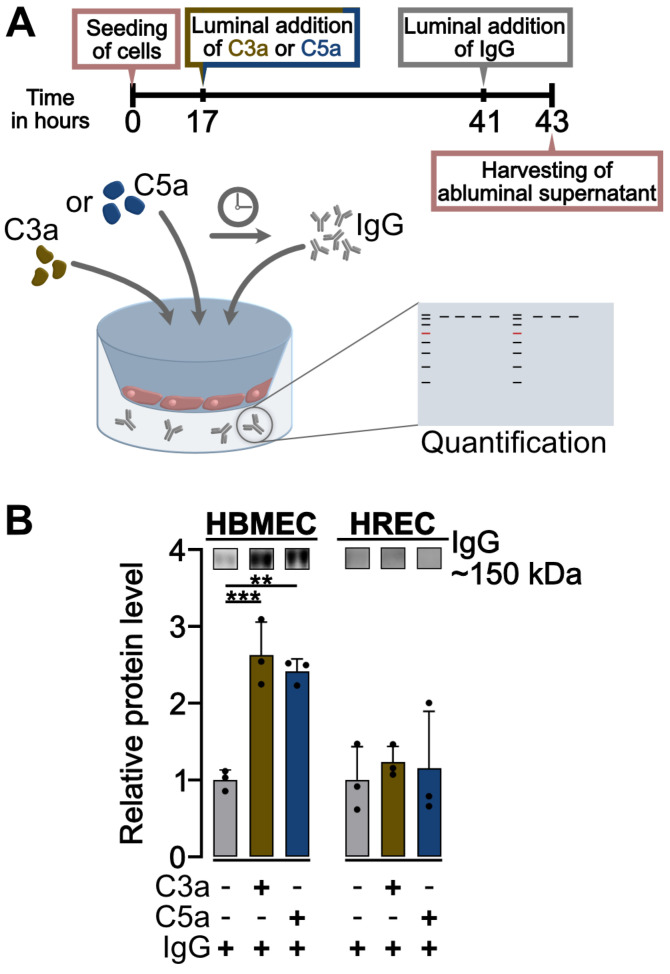
500 nM C3a and C5a increase the transcellular permeability for IgG in HBMEC but not in HREC, reflecting reduced cell–cell contact integrity at 500 nM C3a but not enhanced paracellular resistance at 500 nM C5a in HBMEC. (**A**) Endothelial cells were allowed to settle and form a continuous monolayer 17 h prior to the addition of 500 nM C3a or 500 nM C5a. After 24 h, 100 µg/mL purified IgG was applied luminally, and abluminal supernatants were harvested 2 h later to assess IgG migration through the endothelial barrier using Western blot detection. (**B**) Western blot quantification revealed a significant increase in IgG migration through the endothelial barrier in 500 nM C3a- and 500 nM C5a-treated HBMEC compared to the untreated control. This effect was not observed in HREC, where the IgG migration rate was similar to the untreated control in cells treated with 500 nM C3a and 500 nM C5a. Bar height represents the mean relative protein level for *n* = 3 for all treatment groups in both cell types. Cell types were analyzed as separate datasets. Uncropped Western blots are shown in Appendix A. One-way ANOVA with Dunnett’s T3 *post hoc* test for parametric datasets. ** *p* < 0.01, *** *p* < 0.001.

**Table 1 ijms-25-11240-t001:** In-house designed qRT-PCR primers.

Transcript	Sequence (5′-3′)
*CDH5*	f: GGACCGAGAGAAGCAGGCCA r: TGTGTACTTGGTCTGGGTGAAGA
*C3*	f: AAG AAC CGC TGG GAG GAC CC r: ATT GAG CCA ACG CAC GAC GG
*C3AR1*	f: TTC CGA ATG CAA AGG GGC CG r: ACC ACG GCC ACT CGA AAG GT
*GAPDH*	f: CCCCACCACACTGAATCTCC r: GGTACTTTATTGATGGTACATGACAAG

**Table 2 ijms-25-11240-t002:** QuantiTec PrimerAssays for qRT-PCR.

Transcript	Name	Catalogue Number (Qiagen, Germany)
*C5*	Hs_C5_1_SG	QT00088011
*C5AR1*	Hs_C5R1_1_SG	QT00997766

## Data Availability

The raw data supporting the conclusions of this article will be made available by the authors on request.

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
