# Peer review of "C3a Mediates Endothelial Barrier Disruption in Brain-Derived, but Not Retinal, Human Endothelial Cells"

_ijms, 2024, doi:10.3390/ijms252011240_

Round 1

Reviewer 1 Report

Comments and Suggestions for Authors

This manuscript details experiments on cell cultures of human brain endothelial cells and human retinal endothelial cells exposed to anaphylotoxins C3a and C5a. The authors find that retinal endothelial cell permeability is generally less susceptible to anaphylotoxin treatment than that seen in brain endothelial cells. In addition, in brain endothelial cells, low dose C5a given with higher-dose C5a is protective against the increased cellular permeability. The manuscript is overall well written, and the figures are well presented. Some of the conclusions go beyond the data, and there are a few minor typo-level issues that should be corrected.

- much of the data presented in Figures 2, 4, 5, 6, and 7 relies solely on subjective interpretation of a single representative image from each treatment group. These data would be much more convincing if there had been any quantitative or semi-quantitative analysis (or even a semi-structured qualitative analysis done by blinded investigators – e.g. scoring cell integrity as +, ++, +++). This weakness is mitigated somewhat by the cellular impedance data from RTCA analysis and the IgG translocation data from Figure 8. At the very least, though, this weakness should be acknowledged in the limitations section of the discussion.

- Similarly, some of the data from immunocytochemistry is over-interpreted. For example, line 152-153 concludes that “barrier integrity remained unaltered…” based on histologic data, which is not really able to show this.

- I also recommend avoiding statements such as in line 209 “no alteration in C3 protein detection…” since no quantification was attempted (or at least reported)

- In Figure 6, it appears that C5aR1 expression drops in untreated cells between 2 hours and 24 hours after treatment, in both cell types. Is there any reason this would be expected?

- Line 305 (Figure 6 legend) states that C5aR1 signal “increased in both 500 nM C3a and 500 nM C5a treated HBMEC.”  To my eye this is not the case, and it is difficult to argue it without any kind of quantitative analysis of the photomicrographs.

- The manuscript makes a relatively big deal about a “ratio-dependent” protective effect of co-treating with C3a and C5a. Although there is a ratio between the dosage of the two anaphylotoxins (1:10), it is impossible to conclude that the ratio per se is what leads to the protective effect since only one set of concentrations at this ratio is used. It could just as easily be the specific dose of C5a (e.g. maybe 100 nM C3a + 50 nM C5a would work just as well), or perhaps the absolute difference in doses.

- Heading 3.5 (lines 311-312) equates cell monoculture effects with vascular stability, which really is beyond the scope of the current work. This is acknowledged in the discussion, in the paragraph about limitations of this work, but I recommend more caution about interpreting these results this way in the body of the paper as well.

Minor:

- several of the acronyms are defined in the methods section, but this section is at the end of the manuscript. This should be revised so acronyms are defined when they are first seen

- In the methods section, it appears that superscripts are missing in cell counts (e.g. 1.5 x 105 instead of 1.5 x 105)

Author Response

Dear Reviewer,

Thank you very much for your valuable comments and suggestions. They have been incredibly helpful, and we truly believe that your insights have significantly improved the quality of our manuscript. You will find our responses to your questions and concerns in the attached document.

Reviewer 1

  1. much of the data presented in Figures 2, 4, 5, 6, and 7 relies solely on subjective interpretation of a single representative image from each treatment group. These data would be much more convincing if there had been any quantitative or semi-quantitative analysis (or even a semi-structured qualitative analysis done by blinded investigators – e.g. scoring cell integrity as +, ++, +++). This weakness is mitigated somewhat by the cellular impedance data from RTCA analysis and the IgG translocation data from Figure 8. At the very least, though, this weakness should be acknowledged in the limitations section of the discussion.

We sincerely appreciate your insightful comment. We fully agree that the quantification of the immunocytochemical analysis enhances the overall strength and rigor of the manuscript. In response to your suggestion, we have now quantified all stainings presented in Figures 2, 4, 5, 6, and 7. For the analysis, we used three technical replicates and measured fluorescent intensity, normalizing the results to the untreated control, consistent with the methodology applied throughout our study. We believe that the inclusion of these quantitative data significantly supports the validity of our conclusions. Consequently, we have updated all figure legends and revised the results section accordingly.

  1. Similarly, some of the data from immunocytochemistry is over-interpreted. For example, line 152-153 concludes that “barrier integrity remained unaltered…” based on histologic data, which is not really able to show this.

We appreciate your thoughtful comment and agree that our initial interpretation of the histological data was overstated. After carefully reviewing the relevant sections, we have revised the text to provide a more accurate and balanced interpretation. The corresponding passages have been adjusted accordingly.

Lines 151-152: “Twenty-four hours following treatment, untreated brain and retinal endothelial cells exhibited a continuous monolayer.” – we removed “and functional”

Lines 152-155: “While VE-cadherin signal was significantly increased in 500 nM C3a treated HREC compared to the untreated control (p <.05), this was not the case for HBMEC, where C3a induced a significant reduction in VE-cadherin signal intensity (p <.01) and a disruption of the endothelial monolayer.” – We have removed “barrier integrity remained unaltered” and added the respective results from the histological quantification.

Lines 155-157: “In HBMEC, the VE-cadherin signal integrity significantly increased (p <.001) after 24 hours following the addition of 500 nM C5a and the endothelial monolayer remained unaltered in HREC (Figure 2D).” – We removed “In both cell types, the barrier integrity remained stable…” and replaced it with the according data from the histological quantification.

Lines 390-391: “Immunocytochemistry revealed a homogeneous endothelial monolayer in all treatment groups in HBMEC, 2 hours post-treatment.” – We have removed “revealed a highly functional barrier..”.

  1. I also recommend avoiding statements such as in line 209 “no alteration in C3 protein detection…” since no quantification was attempted (or at least reported)

As a result of the quantitative analysis we performed on the histological data, we have now limited our presentation to the absolute results obtained from this analysis.

We have adjusted the respective section and added the following paragraph:

Lines 215-228: “Immunofluorescent staining of C3 revealed an increase in C3 protein signal in both C3a and C5a treated HBMEC after 2 hours (p <.01) and in HREC after 2 and 24 hours (p <.05) (Figure 4). C3a significantly decreased C3 signal intensity in HBMEC compared to the untreated control after 24 hours (p <.0019) (Figure 4B).

The pattern of C3 protein expression in HBMEC and HREC after 2 hours of anaphylatoxin treatment was consistent with mRNA levels for both C3a and C5a treatments (Figures 3A, 4A). In HREC, the increase in C3 protein levels after 24 hours also corresponded with transcript levels. However, at 24 hours, discrepancies in protein and mRNA became evident in HBMEC (Figures 3B, 4B). The initial rise in C3 protein in HBMEC after 2 hours of anaphylatoxin treatment was followed by a decrease in C3 detection at 24 hours, compared to untreated control, which was not mirrored at the transcript level. This discrepancy in HBMEC due to decreased cellular C3 levels occurring independently of transcription, potentially result through increased C3 secretion or enhanced degradation, leading to partial mismatches between protein and mRNA expression (Figures 3A, B, 4).”

  1. In Figure 6, it appears that C5aR1 expression drops in untreated cells between 2 hours and 24 hours after treatment, in both cell types. Is there any reason this would be expected?

C5AR1 transcript levels remain comparable regardless of treatment or duration. We acknowledge that there is a difference in C5aR1 protein signal intensity between 2 and 24 hours in the histological dataset for the untreated cells. At this stage, we have chosen to focus on the comparison between treatments rather than the two time points, and have therefore not addressed this issue further. Based on your comment, we have adjusted the order of the figures and sorted the data by time point rather than cell type. This way, the restructuring facilitates a clearer and more direct comparison between the two cell types.

However, if we were to interpret the data, several potential explanations could be considered for this result:

  1. Endothelial cells may adapt further to cultivation conditions over time, leading to changes in protein arrangement in the cell. This phenomenon has been well-documented, particularly in retinal pigment epithelial cells (e.g. actin filaments, bestrophin). Consequently, C5aR1 protein might diminish with prolonged culture, especially in cells not exposed to stimuli that would maintain or upregulate its expression.
  2. We have demonstrated that endothelial cells express C5, which may lead to the production of C5a. Upon ligand binding (autocrine regulation), C5aR1 can undergo internalization into endosomes, which could reduce its surface expression and result in a weaker signal during surface-directed immunocytochemical staining. Relocation to intracellular compartments may decrease the detectable signal. This is increasingly recognized, as recent studies indicate that the complement system extends beyond its traditional functions and also operates intracellularly.

  1. Line 305 (Figure 6 legend) states that C5aR1 signal “increased in both 500 nM C3a and 500 nM C5a treated HBMEC.”  To my eye this is not the case, and it is difficult to argue it without any kind of quantitative analysis of the photomicrographs.

Thank you for highlighting this point. We have revised the legend of Figure 6 to include the following description: “After 24 hours, the C5aR1 signal was stable between treatment groups in HBMEC.” (Line 334), which is now also supported by the quantitative data we have added to the figure.

  1. The manuscript makes a relatively big deal about a “ratio-dependent” protective effect of co-treating with C3a and C5a. Although there is a ratio between the dosage of the two anaphylotoxins (1:10), it is impossible to conclude that the ratio per se is what leads to the protective effect since only one set of concentrations at this ratio is used. It could just as easily be the specific dose of C5a (e.g. maybe 100 nM C3a + 50 nM C5a would work just as well), or perhaps the absolute difference in doses.

This is a crucial point. Upon reviewing the results from this perspective, we acknowledge that the rescuing effect could indeed be attributed to a specific dose of C5a or an absolute difference. We have revised the manuscript accordingly and adjusted the relevant text passages. Nevertheless, we maintain that there may be a combinational effect of C3a and C5a, as demonstrated in previous studies (e.g., PMID: 28663750).

Lines 24-26: We have removed “highlighting the therapeutic importance of regulating the C3a and C5a levels in NMOSD patients.”

Lines 68-70: We have changed “This study is the first to demonstrate that C5a attenuated the C3a mediated disruptive effect on the endothelial barrier when C3a and C5a are present at a 10:1 ratio.” to “This study is the first to demonstrate that C5a attenuates the C3a-mediated disruptive effect on the endothelial barrier.”

Lines 345-347: We changed “We determined the ratio-dependent effect on the endothelial barrier integrity using RTCA and immunocytochemical visualization of VE-cadherin.” To “We determined the concentration-dependent effect on the endothelial barrier integrity using RTCA and immunocytochemical visualization of VE-cadherin.”

Lines 369-371: We have changed “After identifying that this effect can be attenuated by a ratio-dependent supplementation with C5a in HBMEC in the RTCA, we investigated if this effect is reflected in VE-cadherin expression and distribution, using immunocytochemistry.” To “After identifying that this effect can be attenuated by a supplementation with 50 nM C5a in HBMEC in the RTCA, we investigated if this effect is reflected in VE-cadherin expression and distribution, using immunocytochemistry.”

Lines 384-386: We have changed “Monitoring of combined anaphylatoxin-treated dnCI changes in HBMEC revealed a C3a-C5a ratio-dependent increase in dnCI, which counteracted the barrier-disruptive effect of C3a.” to “Monitoring of combined anaphylatoxin-treated dnCI changes in HBMEC revealed a C5a dependent increase in dnCI, which counteracted the barrier-disruptive effect of C3a.”

Lines 534-536: We have changed “…one main finding of our study is that a combinational treatment of C3a and C5a at a 10:1 ratio prevents the barrier-disruptive effect mediated by C3a treatment alone in HBMEC.” To “…one main finding of our study is that a combinational treatment of 500 nM C3a and 50 nM C5a prevents the barrier-disruptive effect mediated by C3a treatment alone in HBMEC.”

Line 545: We have changed “Literature is sparse about the overcompensating effect of C5a on C3a and, particularly with regard to the role of a ratio dependent effect.” to “Literature is sparse about the overcompensating effect of C5a on C3a.”

Lines 591-594: We have changed “This disruptive effect on cell level can be prevented by addition of C5a at a 10:1 ratio of C3a:C5a, indicating that the availability of anaphylatoxins at a right ratio is vital for a functional endothelial barrier.” To “This disruptive effect on cellular level can be prevented by addition of C5a, indicating an overcompensating effect of C5a on C3a and that the availability of anaphylatoxins at a right concentration is vital for a functional endothelial barrier.”

  1. Heading 3.5 (lines 311-312) equates cell monoculture effects with vascular stability, which really is beyond the scope of the current work. This is acknowledged in the discussion, in the paragraph about limitations of this work, but I recommend more caution about interpreting these results this way in the body of the paper as well.

Thank you for this comment.We have adjusted the heading from “C5a mitigates the disruptive effect of C3a on endothelial barrier integrity, demonstrating a regulatory role in maintaining vascular stability.” to “C5a mitigates the disruptive effect of C3a, demonstrating a regulatory role in maintaining endothelial barrier integrity”. This way we have excluded the overinterpretation of the data in regard to vascular stability and limited the effect on the endothelial barrier integrity, which we assessed using RTCA and histological data.

  1. several of the acronyms are defined in the methods section, but this section is at the end of the manuscript. This should be revised so acronyms are defined when they are first seen

Thank you for pointing out this discrepancy. We have revised the manuscript accordingly and introduced the acronyms when they were first used in the text.

  1. In the methods section, it appears that superscripts are missing in cell counts (e.g. 1.5 x 105 instead of 1.5 x 105)

Thank you for bringing this to our attention. We have included the superscripts as needed.

Reviewer 2 Report

Comments and Suggestions for Authors

The paper by dr. Wolf and colleagues reports that anaphylatoxins C3a and C5a differently affect brain and retinal vasculature, with the retinal endothelial barrier being more stable under anaphylatoxin-induced stress compared to that of the brain. These observations may contribute to clarify the role of C3a and C5a, and of their ratio, in the pathogenesis of NMOSD. Although the data, overall, confirm the prediction, some adjustments are needed.

1.      Lines 55-64: This paragraph reports some differences between brain and retinal microvasculature, which is important to introduce the point that brain and retinal endothelial cells may have different responses in an NMOSD setting. However, in lines 60-63 it is said that the mechanisms of neurovascular coupling are different in brain and retina, mainly because the retina has specialized glial cells such as the Müller cells, and I don’t think this is correct. Indeed, ref [27] describes neurovascular coupling in the retina, but it does not discuss any difference with the brain. Retinal Müller cells contribute to the neurovascular coupling in the retina as the astrocytes participate to this function in the brain, to the point that observations of retinal neurovascular coupling have been proposed as tools to investigate the brain function (see for instance https://doi.org/10.3389/fendo.2022.1014287).

2.      Lines 67-73: Usually, I don’t like anticipating the results of the study at the end of the introduction, but this is a matter of personal taste, I guess. In any case, the two conclusions in lines 69-71 and 71-73 are difficult to interpret (why are they interesting?) at this point without an adequate discussion.

3.      Line 85: Abbreviation RTCA should be defined here and not on line 565. Please revise all abbreviations (for instance, RTCA should be used in place of “real-time cell analysis” on line 571).

4.      Line 91: Please define dnCI.

5.      Figure 2: To better match the description of the data in the text, I would suggest organizing panel C with HBMEC and HREC at 2h and panel D with HBMEC and HREC at 24h. You may consider this also for the other figures.

6.      Line 171: Paragraph number should be 2.2. Please also revise the other paragraphs.

7.      Figure 4: The immunofluorescence patterns of C3a range from spot-like to intracellularly diffused. On line 180, it is said that the immunostaining was used to screen for increased C3a presence; however, in the text referring to fig.4 there is no mention of different amounts of C3a, but only of different intracellular distribution patterns. In any case, if you want to evaluate possible differences in C3a content in the cells using immunofluorescence, a proper quantitative image analysis should be performed.

8.      Line 225: The immunofluorescence patterns of C3 (no alterations, line 209) are not in line with C3 gene expression (increased C3 gene expression in HREC, line 185).

9.      Figure 5 and related text: Observations of “signal intensity” are not of great value without a quantitative analysis. In addition, it seems that the C5 distribution pattern in HREC changes with time, from 2 to 24 hours, even in the absence of any treatment.

10.  Line 290-291: ”… in untreated cells 2 hours post-treatment …” seems quite odd.

11.  Paragraph 3.4: As noted in point 7, “signal intensity” is not a reliable index of the amount of the protein under investigation. The evaluation of signal intensity is based mainly on the appearance of the higher magnification inserts in the immunofluorescence figures; therefore, this evaluation depends on the area chosen for magnification. Just to make an example, in figure 6C take the second panel at 24h: if you move the magnified area just above the one selected here, you get a completely different result. If you want to get reliable information from these experiments, you should perform quantitative image analysis in multiple fields from repeated experiments. Or if your aim is to quantify relative protein levels, you may use Western blotting.

12.  Lines 313-314: This sentence aims to summarize the findings reported in figure 1 and 3. However figure 3 does not seem to indicate any barrier disruptive or enhancing effect: it just represents the expression of C3 and C3AR1. In addition, while figure 1 shows barrier disruptive effects of C3a at the highest concentration and barrier-enhancing effect of C5a in HBMEC, in HREC both C3a and C5a seem to have strong enhancing effects but only at low concentrations, which is not exactly what is summarized in lines 313-314.

13.  Lines 414-416: As noted in the previous point, both C3a and C5a (at low concentrations) enhance the barrier in HREC. Why is this effect of C3a disregarded?

14.  Lines 467-485: This part of the discussion considers that C3a and C5a upregulate C3 and C5 gene expression in HREC and this is interpreted as an indication of complement-mediated protection of retinal vessels. Then, C3AR1 expression is considered (you should point out that only with C5a and only at 2h the effect is seen in HBMEC), but an interpretation of these data is missing: what was expected from these experiments and what do the results indicate, overall?

15.  Line 483, “… similar to our findings in HBMEC”: C5AR1 was not upregulated in HBMEC (Fig. 6A). Then a discussion of the C5AR1 data is missing: why was that analysis performed? What are the results? What do they mean?

16.  Lines 486-497: As noted in point 2, I don’t see a rationale to investigate the effects of combinations of C2 and C5: what was the expectation? And what do the results indicate? Why are these data important? The conclusion that “this indicates a differential effect of both anaphylatoxins in regulating intracellular calcium levels and associated paracellular barrier integrity” is not sufficient.

17.  A consideration of the immunofluorescence data is completely missing from the discussion. So, why were these experiments done?

Comments on the Quality of English Language

I suggest moderate editing of English language.

Author Response

Dear Reviewer,

Thank you so much for your valuable feedback. Your insights have significantly helped us improve our manuscript, and we truly appreciate the time you took to review it. Our responses to your comments can be found in the attached document.

Reviewer 2

  1. Lines 55-64: This paragraph reports some differences between brain and retinal microvasculature, which is important to introduce the point that brain and retinal endothelial cells may have different responses in an NMOSD setting. However, in lines 60-63 it is said that the mechanisms of neurovascular coupling are different in brain and retina, mainly because the retina has specialized glial cells such as the Müller cells, and I don’t think this is correct. Indeed, ref [27] describes neurovascular coupling in the retina, but it does not discuss any difference with the brain. Retinal Müller cells contribute to the neurovascular coupling in the retina as the astrocytes participate to this function in the brain, to the point that observations of retinal neurovascular coupling have been proposed as tools to investigate the brain function (see for instance https://doi.org/10.3389/fendo.2022.1014287).

    Dear Reviewer, thank you for your valuable comment regarding neurovascular coupling in our introduction. We have reviewed the literature and gained further insights on the topic. Accordingly, we have revised the relevant section and removed the respective sentence.

  1. Lines 67-73: Usually, I don’t like anticipating the results of the study at the end of the introduction, but this is a matter of personal taste, I guess. In any case, the two conclusions in lines 69-71 and 71-73 are difficult to interpret (why are they interesting?) at this point without an adequate discussion.

      Thank you for expressing your preference regarding the summary of the results at the end of the introduction. While we personally favor including a brief summary at this point, we acknowledge that the last two sentences were difficult to interpret. Therefore, we have adjusted the respective paragraph.

      Lines 68-70: We have changed “Interestingly, C3 and C5 gene expression increased following C3a and C5a treatment in retinal, but not in brain-derived endothelial cells. This study is the first to demonstrate that C5a attenuates the C3a-mediated disruptive effect in the endothelial barrier when C3a and C5a are present at a 10:1 ratio.” To “This study is the first to demonstrate that C5a attenuates the C3a-mediated disruptive effect on the endothelial barrier.”

  1. Line 85: Abbreviation RTCA should be defined here and not on line 565. Please revise all abbreviations (for instance, RTCA should be used in place of “real-time cell analysis” on line 571).

Thank you for pointing out this discrepancy. We have made the necessary adjustments to the manuscript and introduced the abbreviations when they were first used in the text.

  1. Line 91: Please define dnCI.

      Thank you for bringing this to our attention. We have defined dnCI and included the following sentence in the methods section::

Lines 653-654: “The CI of each curve was double normalized (dnCI) to its respective CI value measured before the addition of anaphylatoxin treatment and to the untreated control.”

  1. Figure 2: To better match the description of the data in the text, I would suggest organizing panel C with HBMEC and HREC at 2h and panel D with HBMEC and HREC at 24h. You may consider this also for the other figures.

      Thank you for this very constructive feedback. We have reorganized the figures accordingly, and we agree that this restructuring facilitates a clearer and more direct comparison between the two cell types.

  1. Line 171: Paragraph number should be 2.2. Please also revise the other paragraphs.

      Thank you for pointing out this discrepancy in the numbering of our manuscript. We have adjusted the paragraph number in the results section accordingly.

  1. Figure 4: The immunofluorescence patterns of C3a range from spot-like to intracellularly diffused. On line 180, it is said that the immunostaining was used to screen for increased C3a presence; however, in the text referring to fig.4 there is no mention of different amounts of C3a, but only of different intracellular distribution patterns. In any case, if you want to evaluate possible differences in C3a content in the cells using immunofluorescence, a proper quantitative image analysis should be performed.

Thank you very much for your valuable comment. We completely agree that quantifying the immunocytochemical analysis enhances the message of our manuscript, not only for Figure 4 but also for Figures 2, 5, 6, and 7. Consequently, we have quantified all stainings presented in these figures. For the analysis, we utilized three technical replicates and assessed fluorescent intensity, normalizing the results to the untreated control, consistent with our other data. We now believe that our statements are well-supported by this quantitative data. We have accordingly updated all figure legends and revised the results section. However, we retained the description of the cellular distribution pattern and included a description of the quantitative data.

Lines 255-256 (Figure legend 4): “HBMEC and HREC C3a signal intensity maintained stable across treatment groups and exhibited a spot-wise pattern across all treatment groups after 2 hours.”

Lines 262-263 (Figure legend 4): “C3 signal intensity increased in C3a and C5a treated HREC after 24 hours.” (was added for HREC)

Lines 236-238 (main text): “The C3a protein signal intensity remained stable regardless of treatment and time point in HBMEC.” (was added)

Lines 242-244 (main text): “and showed a significantly increased C3a signal compared to the untreated control (untreated vs. 500 nM C3a: p .<05; untreated vs. 500 nM C5a: p <.01)” (was added)

  1. Line 225: The immunofluorescence patterns of C3 (no alterations, line 209) are not in line with C3 gene expression (increased C3 gene expression in HREC, line 185).

We have adjusted the relevant section for a more precise description of the results:

Lines 215-228: “Immunofluorescent staining of C3 revealed an increase in C3 protein signal in both C3a and C5a treated HBMEC after 2 hours (p <.01) and in HREC after 2 and 24 hours (p <.05) (Figure 4). C3a significantly decreased C3 signal intensity in HBMEC compared to the untreated control after 24 hours (p <.0019) (Figure 4B).

The pattern of C3 protein expression in HBMEC and HREC after 2 hours of anaphylatoxin treatment was consistent with mRNA levels for both C3a and C5a treatments (Figures 3A, 4A). In HREC, the increase in C3 protein levels after 24 hours also corresponded with transcript levels. However, at 24 hours, discrepancies in protein and mRNA became evident in HBMEC (Figures 3B, 4B). The initial rise in C3 protein in HBMEC after 2 hours of anaphylatoxin treatment was followed by a decrease in C3 detection at 24 hours, compared to untreated control, which was not mirrored at the transcript level. This discrepancy in HBMEC due to decreased cellular C3 levels occurring independently of transcription, potentially result through increased C3 secretion or enhanced degradation, leading to partial mismatches between protein and mRNA expression (Figures 3A, B, 4).”

  1. Figure 5 and related text: Observations of “signal intensity” are not of great value without a quantitative analysis. In addition, it seems that the C5 distribution pattern in HREC changes with time, from 2 to 24 hours, even in the absence of any treatment.

Thank you for highlighting this point. Indeed, a quantitative analysis of the data enhances the rigor of the entire study. We have conducted a quantitative analysis of the C5 immunocytochemical data, along with all other relevant data (see response to point 7), and have adjusted the figure legend to incorporate the results of our quantitative analysis.

We have also noticed the different distribution pattern of C5 between 2 and 24 hours in HREC. There might be different reasons for this, however non of which is currently proven with experiments. We observed no changes in mRNA expression between 2 and 24 hours of cultivation time. During prolonged cultivation, metabolic changes may occur in cells, which cannot be ruled out. This might involve a downregulation of pathways involved in complement protein synthesis or a shift in cellular priorities, leading to lower C5 levels.

  1. Line 290-291: ”… in untreated cells 2 hours post-treatment …” seems quite odd.

      Lines 320-321: We have exchanged “In HBMEC, we observed a stronger C5aR1 signal in untreated cells 2 hours post-treatment than in cells treated with…” with “After 2 hours, we observed a stronger C5aR1 signal in untreated HBMEC than in cells treated with 500 nM C3a or 500 nM C5a (p <.05).”

  1. Paragraph 3.4: As noted in point 7, “signal intensity” is not a reliable index of the amount of the protein under investigation. The evaluation of signal intensity is based mainly on the appearance of the higher magnification inserts in the immunofluorescence figures; therefore, this evaluation depends on the area chosen for magnification. Just to make an example, in figure 6C take the second panel at 24h: if you move the magnified area just above the one selected here, you get a completely different result. If you want to get reliable information from these experiments, you should perform quantitative image analysis in multiple fields from repeated experiments. Or if your aim is to quantify relative protein levels, you may use Western blotting.

We also conducted a quantitative analysis for Figure 6 and have incorporated it into the dataset. Both the figure legend and the results description have been adjusted accordingly.

  1. Lines 313-314: This sentence aims to summarize the findings reported in figure 1 and 3. However figure 3 does not seem to indicate any barrier disruptive or enhancing effect: it just represents the expression of C3 and C3AR1. In addition, while figure 1 shows barrier disruptive effects of C3a at the highest concentration and barrier-enhancing effect of C5a in HBMEC, in HREC both C3a and C5a seem to have strong enhancing effects but only at low concentrations, which is not exactly what is summarized in lines 313-314.

Thank you very much for pointing out this discrepancy. It was indeed a typo, and the sentence should reference Figures 1 and 2. We have corrected the reference. Since the sentence primarily serves as an introductory line to the subsequent results section concerning the combinational effect of C3a and C5a, we have chosen not to include the enhancing effect of C3a in HREC. Our main focus is on the difference in barrier disruption between the two cell types following C3a stimulation and how C5a can counteract this effect. Therefore, it serves more as a reference for the reader to locate the respective results we cited, rather than as a comprehensive summary of the figures.

  1. Lines 414-416: As noted in the previous point, both C3a and C5a (at low concentrations) enhance the barrier in HREC. Why is this effect of C3a disregarded?

      The primary focus of the manuscript is the differential stability of both endothelial linings in the context of systemic complement-related stress and the potential opening of the endothelial barrier to facilitate the migration of pathological IgG from the blood into the central nervous system or the retina. Therefore, we concentrated on the key difference between the two cell types that addresses our main question—the disruptive effect of C3a on HBMEC, but not on HREC. However, we acknowledge that it is also an important finding to demonstrate that C3a can enhance the paracellular endothelial barrier in HREC. Consequently, we have revised the relevant sentence to include the barrier-enhancing effect of low levels of C3a in HREC.

      Lines 448-450: We changed “We report a C3a dose-dependent decrease in paracellular impedance in HBMEC, but not in HREC, while low levels of C5a stimulation exerted a barrier-enhancing effect in both endothelial cell types.” To “We report a C3a mediated decrease in paracellular impedance in HBMEC, whereas low levels of C3a increased transendothelial resistance in HREC. Low levels of C5a stimulation exerted a barrier-enhancing effect in both endothelial cell types.”

  1. Lines 467-485: This part of the discussion considers that C3a and C5a upregulate C3 and C5 gene expression in HREC and this is interpreted as an indication of complement-mediated protection of retinal vessels. Then, C3AR1 expression is considered (you should point out that only with C5a and only at 2h the effect is seen in HBMEC), but an interpretation of these data is missing: what was expected from these experiments and what do the results indicate, overall?

The analysis was performed to explore potential early signaling mechanisms involving the C3a-C3aR and C5a-C5aR pathways. Given the pivotal role of anaphylatoxins in immune modulation and endothelial function, we aimed to investigate whether these  protein-activation product-receptor axes are differentially regulated in these cell types in response to anaphylatoxins. The results, showing a downregulation of C5aR (following 2 h after C5a treatment) and a potential compensatory upregulation of C3aR at the transcript level in HBMECs, suggest that cross-talk between these pathways may contribute to early cellular responses. In contrast, the absence of similar effects in HRECs indicates that these cells might possess a different regulatory mechanism or may be less susceptible to early complement-related signaling changes. These findings provide a foundation for further investigation into the distinct roles and susceptibilities of endothelial cells in complement receptor-mediated processes.  

We included the involvement of a 2 hour C5a stimulation into the discussion.

Lines 512-514: “While we determined a regulation of C3 and C5 gene transcription in HREC, we observed a significant increase in C3AR1 expression following 2 hours of C5a stimulation in HBMEC, but not HREC.”

      We have expanded our discussion regarding the regulation of C3AR1 in HBMEC, but not in HREC. We have added the following paragraph:

Lines 521-531: “This indicates a form of cross-talk between the C5a-C5aR and C3a-C3aR pathways. In fact, we report that after 2 hours of C3a and C5a treatment, a reduction in C5aR1 protein detection is observed in HBMEC. This C5aR1 downregulation is accompanied by a potential compensatory upregulation of C3aR at the transcript level in HBMEC. While this upregulation shows only a trend for C3a, it is statistically significant for C5a, indicating a transient increase in C3aR transcription. This may initiate an amplification of the C3a-mediated effect within brain endothelial cells. In contrast, HREC did not exhibit early adjustments in the expression of anaphylatoxin receptors, either at the protein or transcript level. As a possible consequence, the intact HREC barrier may provide initial insights into the differential mechanisms underlying endothelial cell type-specific susceptibility to systemic complement-related stress. “

  1. Line 483, “… similar to our findings in HBMEC”: C5AR1 was not upregulated in HBMEC (Fig. 6A). Then a discussion of the C5AR1 data is missing: why was that analysis performed? What are the results? What do they mean?

      We performed analysis of C5aR1 transcript and protein expression to explore potential early mechanisms involving both the C3a-C3aR and the C5a-C5aR pathways following interaction with C3a and C5a.

We have adjusted the order of the sentence:

Lines 517-519: “Specifically in HUVECs, exposure to 100 nM C3a and 100 nM C5a resulted in an upregulation of C5AR1 and, similar to our findings in HBMEC, C3AR1 gene expression [19].”

We have included the following sentences in the discussion to address C5AR1 gene expression and placed it within the context of existing literature:

Lines 514-519: “C5AR1 gene expression was stable in both cell types regardless of treatment or duration. Endothelial anaphylatoxin receptor expression has been described before in HUVEC, HBMEC, mouse dermal microvascular endothelial cells and choroidal endothelial cells [19,20,37,50]. Specifically in HUVECs, exposure to 100 nM C3a and 100 nM C5a resulted in an upregulation of C5AR1 and, similar to our findings in HBMEC, C3AR1 gene expression [19].”

  1. Lines 486-497: As noted in point 2, I don’t see a rationale to investigate the effects of combinations of C2 and C5: what was the expectation? And what do the results indicate? Why are these data important? The conclusion that “this indicates a differential effect of both anaphylatoxins in regulating intracellular calcium levels and associated paracellular barrier integrity” is not sufficient.

Thank you for critically reviewing our data. We believe you are referring to the combinational treatment of C3a and C5a at different ratios. There are several reasons why investigating both the combinational effect and the impact of each anaphylatoxin on the expression of the other's precursor protein and receptor is of interest:

  1. Structural Homology: C3a and C5a are structurally highly homologous and belong to the anaphylatoxin protein family. Specifically, both proteins possess a small α-helical structure stabilized by disulfide bonds. They feature a C-terminal arginine residue, which is crucial for their anaphylatoxin activity; removal of this residue diminishes their receptor activation. Additionally, their unique globular shape is maintained by three disulfide bonds, essential for receptor binding. These similarities suggest that the C3a-C3aR and C5a-C5aR pathways may cross-talk or exhibit a combinational effect, as demonstrated in retinal pigment epithelial cells (PMID: 28663750). C3a and C5a can exhibit opposing effects, with C3a often showing anti-inflammatory properties, while C5a primarily acts as a strong pro-inflammatory anaphylatoxin. These contrasting, as well as synergistic, effects between C3a and C5a have been well-documented, particularly in studies on kidney diseases, where their interaction can vary depending on the specific pathological context (10.4049/jimmunol.1403068; 10.1159/0005382419).

  2. Physiological Context: C3a is cleaved from its precursor protein before C5a and is generally more abundant systemically. However, conditions such as sepsis and various autoimmune diseases can lead to dysregulation of the complement system. In certain states, complement activation may accelerate, causing differences in the systemic ratios of C3a and C5a. In NMOSD, it has been shown that both anaphylatoxins are dysregulated. We have included the following sentence:

Lines 43-44: “Complement activation significantly contributes to NMOSD pathophysiology, and systemic C3a and C5a levels are altered in NMOSD patients [12–16].”

  1. Therapeutic Implications: Several therapies related to the complement system are now available. Specifically, the approval of Eculizumab, a monoclonal C5 inhibitor, has demonstrated significant efficacy in reducing relapse rates in AQP4-IgG seropositive NMOSD patients. This suggests a systemic dysregulation of complement anaphylatoxins.

Given that both anaphylatoxins are present simultaneously in physiological conditions, share structural homology, have ratios that can be influenced by autoimmune diseases, and considering the positive effect of the complement therapeutic Eculizumab in preventing NMOSD relapses, we chose to explore a largely underrepresented topic in current complement research: the influence of the systemic ratio of C3a and C5a on each other's pathways.

We have added the following sentences to the discussion to provide an alternative perspective on this topic:

Lines 536-544: “While elevated C5a levels reduced the disruptive effect of C3a on HBMEC, only the lowest concentration of C5a significantly increased paracellular impedance. This suggests that small amounts of C5a help maintain the integrity of endothelial cells, preventing them from becoming 'leaky.' However, when C5a levels become too high, the barrier becomes compromised again. This highlights the importance of not completely blocking C5a, but rather maintaining it at low levels, as achieved with the monoclonal C5 inhibitor Eculizumab in NMOSD [52]. A balance must be maintained, much like the natural regulation of the complement cascade, where C3a is produced in significantly higher amounts early in the cascade, while C5a appears later in much smaller quantities.”

  1. A consideration of the immunofluorescence data is completely missing from the discussion. So, why were these experiments done?

      In general, we personally prefer to discuss main findings and potential mechanisms in those regards rather than single figures. Hence, we have discussed e.g. the variation in VE-cadherin signal between both cell types by introducing a general mechanism that might be responsible for a differential regulation between both cell types. We have included the following paragraph:

Lines 467-479: “Mechanistically, the induction of endothelial hyperpermeability is still under investigation. A likely event involved in endothelial barrier breakdown is the release of calcium from the endoplasmatic reticulum and the subsequent increase in intracellular calcium levels, e.g. following C3a-C3aR interaction. Elevated calcium levels can cause a downregulation of VE-cadherin by increasing the expression of endothelial activation marker vascular cell adhesion molecule-1 (VCAM-1) [34]. Besides the upregulation of VCAM-1, the activation of protease activating receptors (PARs), specifically PAR1, by anaphylatoxins could be another potential, barrier disruptive pathway that has not yet been investigated. Wang et al. previously reported that anaphylatoxin C4a, which is structurally highly similar to C3a and C5a, can bind to PAR1, leading to enhanced endothelial permeability [39]. PAR1 activation leads to the activation of numerous intracellular signalling pathways, ultimately resulting in nitric oxide production, which modulates phosphorylation and thereby downstream internalization of VE-cadherin [40].“

Additionally, we have discussed differences between both tissues that might explain differential reactions to anaphylatoxic stress, which of course includes phenotypic, transcriptomic and proteomic data sets.

Lines 480-500: “The stability of HREC against C3a indicates a tissue-specific protection from complement anaphylatoxin-mediated stress, and thereby potentially from paracellular autoantibody transmigration through the inner blood-retina-barrier. A distinct response of the endothelial vasculature from different tissues to stressors can be expected, as organ-specific characteristics in endothelial cells are induced and maintained by the surrounding microenvironment, resulting in a high inter-tissue heterogeneity [23]. Even within one tissue, contour-based 3D image visualization and quantification revealed a high heterogeneity in tight junction protein Claudin-5 protein expression within the murine central nervous system microvasculature [41]. Between different tissues, this becomes specifically evident as the gene transcriptome differs in brain and retinal microvessels in rats [42]. A distinct response to different forms of systemic stress is therefore unsurprising. For example, bovine retinal endothelial cells are more susceptible to glucose-induced oxidative stress than brain-derived endothelial cells, whereas we recently reported a higher susceptibility to oxygen-induced oxidative stress in brain-derived endothelial cells compared to retinal endothelial cells [28,43]. Similarly, alternative complement pathway protein expression, activation and regulation vary highly between human glomerular and brain-derived microvascular endothelial cells, highlighting once more the high inter-tissue heterogeneity in endothelial cells [44]. It remains difficult to discuss the specific influence of C3a and C5a on the endothelium of different tissues due to the lack of comparative studies in this field and therefore the absolute comparison of abundance and availability of anaphylatoxin receptors in specific endothelial cell types.“

In response to your comment, we have added the following statements related to complement protein regulation following anaphylatoxin treatment in the discussion:

Lines 503-505: “After exposure to C3a and C5a we observed an upregulation in C3 and C5 transcript and protein detection in HREC, but only for C3 protein in HBMEC.”

Lines 512-514: “While we determined a regulation of C3 and C5 expression in HREC, we observed a significant increase in C3AR1 expression following 2 hours of C5a stimulation in HBMEC, but not HREC.”

Lines 521-531: “This indicates a form of cross-talk between the C5a-C5aR and C3a-C3aR pathways. In fact, we report that after 2 hours of C3a and C5a treatment, a reduction in C5aR1 protein detection is observed in HBMEC. This C5aR1 downregulation is accompanied by a potential compensatory upregulation of C3aR at the transcript level in HBMEC. While this upregulation shows only a trend for C3a, it is statistically significant for C5a, indicating a transient increase in C3aR transcription. This may initiate an amplification of the C3a-mediated effect within brain endothelial cells. In contrast, HREC did not exhibit early adjustments in the expression of anaphylatoxin receptors, either at the protein or transcript level. As a possible consequence, the intact HREC barrier may provide initial insights into the differential mechanisms underlying endothelial cell type-specific susceptibility to systemic complement-related stress. “

Reviewer 3 Report

Comments and Suggestions for Authors

In the submitted manuscript, Wolf et al. used primary endothelial cells from the retina and brain to compare the stability of the endothelial barrier under stress induced by C5a and C3a through a series of in vitro experiments. This study is unique, as I have not encountered any research comparing endothelial cell responses between the brain and retina in the context of NMOSD. The authors reported that the retinal endothelial barrier remains stable under C3a stimulation, while brain endothelial cells show increased permeability at high C3a concentrations. The authors also found that C3 and C5 gene expression increased in retinal but not brain endothelial cells after treatment with C3a and C5a. Additionally, they showed that C5a mitigates the disruptive effect of C3a on the endothelial barrier when the two are present at a 10:1 ratio. This study holds potential and is valuable to the field.

Broad comments:

1.     The study focuses solely on 2D culture of endothelial cells (in vitro), which does not fully capture the tissue-specific microenvironment. Further research using 3D culture systems or in vivo models is necessary to better simulate physiological conditions. This limitation has been acknowledged in the discussion section.

2.     To gain a deeper understanding of the mechanisms underlying the differential treatment responses between retinal and brain  endothelial cells, transcriptomic and proteomic analyses could be valuable.

3.     Additionally, the cell viability data, especially for the 500 nM treatment, should be presented to confirm that this dose is not toxic to the cells.

4.     It is crucial to specify the passage number of cells, particularly for primary cells, to ensure the retention of their proper characteristics. The authors should clearly state the passage number used in each experiment and confirm that the endothelial properties of the cells were maintained throughout the study.

Other specific query/comment/suggestions:

·      “Figure 1: Anaphylatoxins C3a and C5a differentially regulate endothelial monoculture dnCI in a 117 dose-dependent manner. (A, B) Monitoring of dnCI changes in response to 50 nM, 100 nM and 500 118 nM of anaphylatoxins C3a and C5a revealed a transient decrease in dnCI by 500 nM C5a after two 119 hours of treatment in HBMEC. 24 hours after anaphylatoxin stimulation, 500 nM C3a significantly 120 decreased the paracellular resistance in HBMEC……..”

For n=6 or 9, were the cells used all from the same passage, or were they from different passages?

·      “Both anaphylatoxins were diluted in the respec- 563

tive culture medium. Treatment was administered at concentrations of 50 nM, 100 nM and 564

500 nM for real time cell analysis (RTCA) and 500 nM for immunocytochemical staining 565

and qRT-PCR samples. Unaltered culture medium was added to the untreated control. 566”

What were the criteria for dose selection? Why was only the 500 nM dose used for immunostaining?

·      “Monitoring of dnCI changes in response to 50 nM, 100 nM and 500 118 nM of anaphylatoxins C3a and C5a revealed a transient decrease in dnCI by 500 nM C5a after two 119hours of treatment in HBMEC. 24 hours after anaphylatoxin stimulation, 500 nM C3a significantly”

Was cell viability assessed in response to the anaphylatoxins? Was an MTT assay or another viability test performed to confirm the effects of 50 nM, 100 nM, and 500 nM treatments?

·      “Figure 1: Anaphylatoxins C3a and C5a differentially regulate endothelial monoculture dnCI in a 117 dose-dependent manner”

Why was the analysis limited to only 24 hours when the xCELLigence assay was run for more than 50 hours? Additionally, was fresh media and C3a replenished during this time, considering their short half-life? Would extending the time points provide more comprehensive insights?

·      “Figure 3: Anaphylatoxin treatment caused increased C3 expression in HREC and increased C3AR1 199 expression in HBMEC. (A, B, C, D) qRT-PCR analysis revealed no effect of anaphylatoxin treatment 200 on C3 expression in HBMEC, but disclosed an increase in C3 expression in 500 nM C5a treated HREC 201 2 hours post-treatment and a 500 nM C3a and 500 nM C5a mediated increase in C3 gene expression 202 24 hours post-treatment in HREC. Two hours exposure to 500 nM C5a increased C3AR1 expression…”

What is the sample size (n) used in this analysis? In panel B, there appears to be a high standard deviation. Can the authors provide justification for this? Were the data collected from four separate cultures considered technical replicates? What passage number was used for the cells?"

·      “Figure 6: C5AR1 transcript expression remains unchanged, but protein detection changes under an- 301aphylatoxic stress. (A, B) qRT-PCR analysis showed no difference in C5AR1 gene expression be- 302 tween untreated and C3a or C5a treated HBMEC and HREC. (C) 500 nM C3a and 500 nM C5a 30….”

Why are untreated HREC showing different C5AR1 levels at 2 hours compared to 24 hours?

Why does the 2-hour time point show higher expression in both cell types in the untreated control group compared to the 24-hour time point?"

·      “….. 500 nM C3a and 500 nM C5a 303 treatment reduced C5aR1 protein signal in HBMEC after 2 hours of treatment. After 24 hours, the 304 C5aR1 signal decreased in untreated cells but increased in both 500 nM C3a and 500 nM C5a treated 305

HBMEC…..”

The representative image does not accurately reflect the stated observations. Better-quality images that support the authors' claims should be included. Additionally, performing quantification using ImageJ would enhance the analysis and understanding of the results. Highlighting just one location in the whole slide may not adequately represent the overall behavior of the cells.

·      “E) Immunocytochemistry revealed a highly functional barrier in HBMEC treated with 500 nM C3a 359……..

+ 50 nM C5a, 2 hours post-treatment.”

A better representative image is required. From Figure 7E, it appears that the combination treatment shows a stronger Cadherin barrier compared to the untreated control group. Can the authors provide justification for this observation? Additionally, how was the combinatorial dose composition (10:1) for the treatment selected?

Author Response

Dear Reviewer,

We greatly appreciate your constructive comments and the time you invested in reviewing our manuscript. Your suggestions have made a real difference, and we believe the manuscript is much stronger thanks to your input. Please find our responses in the enclosed document.

REVIEWER 3

  1. The study focuses solely on 2D culture of endothelial cells (in vitro), which does not fully capture the tissue-specific microenvironment. Further research using 3D culture systems or in vivo models is necessary to better simulate physiological conditions. This limitation has been acknowledged in the discussion section.

Thank you for this valuable comment regarding the use of an endothelial monoculture in our experiments. We completely agree that employing a 3D culture system would more accurately mimic in vivo physiological conditions. In this exploratory study, we focused on the differences in endothelial monocultures to establish a foundational understanding of the distinctions between the two endothelial cell types in response to anaphylatoxic stress. For future studies, it would be highly beneficial to include additional cell types from the blood-brain barrier or the inner blood-retina barrier to better replicate the in vivo environment, such as pericytes, astrocytes, or Müller cells. We have included the following statement in the future outlook of the study:

Lines 603-606: “Furthermore, replicating these treatments using a 3D cell culture system that incorporates barrier-specific cell types, such as pericytes, astrocytes, or Müller cells, will enhance endothelial characteristics and reinforce the findings of this exploratory study.”

2.     To gain a deeper understanding of the mechanisms underlying the differential treatment responses between retinal and brain  endothelial cells, transcriptomic and proteomic analyses could be valuable.

We completely agree that gaining insights into the mechanisms underlying the response to anaphylatoxic stress would be very interesting. Currently, this investigation unfortunately falls beyond the scope of our work; however, we would certainly be interested in exploring the differences between the two endothelial cell types in this context. In addition to examining the response mechanisms to C3a and C5a-mediated stress, it would also be worthwhile to investigate complement inhibitory surface proteins, as these may provide further explanations for tissue-specific protection against systemic complement activation. Based on this, we have included the following statements as a future outlook for the study:

Lines 597-603: “As a future outlook it would be sensible to include transcriptomic and proteomic analyses to gain a deeper understanding of the cell specific mechanisms underlying the response to anaphylatoxic stress. Gene transcription analysis of complement inhibitory surface proteins could be a future approach to assess differential expression in both endothelial cell types, which might determine whether there is a tissue-specific protection from complement-mediated bystander injury.”

3.     Additionally, the cell viability data, especially for the 500 nM treatment, should be presented to confirm that this dose is not toxic to the cells.

Indeed, additional analysis of cell viability following treatment with 500 nM C3a or C5a could provide further support for our findings. Currently, we have assessed the viability of the endothelial cells using two methods. First, we employed real-time cell analysis to evaluate the barrier integrity of the endothelial monolayer after treatment with various concentrations of anaphylatoxins. This approach allowed us to assess the functionality of the endothelial cells in a controlled environment.

While we acknowledge that the decrease in dnCI in HBMEC treated with 500 nM C3a may suggest potential toxicity, we confirmed cell viability analyzing the endothelial-specific marker VE-Cadherin to ensure a stable phenotype. CDH5 gene expression remained stable across all treatment groups, and histological data indicated that the cells maintained viability throughout the treatment duration (Figure 2). A pertinent example of CDH5 gene expression and VE-Cadherin protein distribution during a toxic treatment can be observed in our recent publication regarding exposure to high levels of oxidative stress (e.g., 500 or 2500 µM): 10.1016/j.expneurol.2024.114919.

We hope you will agree that the viability of both cell types has been sufficiently demonstrated using three different techniques: real-time evaluation of endothelial functionality, gene expression analysis of an endothelial-specific marker, and assessment of endothelial phenotype integrity.

  1.    It is crucial to specify the passage number of cells, particularly for primary cells, to ensure the retention of their proper characteristics. The authors should clearly state the passage number used in each experiment and confirm that the endothelial properties of the cells were maintained throughout the study.
    Thank you for this valuable comment. We completely agree that passage number is crucial, especially when working with primary cells. We have included the following statement in the methods section:

Lines 622-623: “Passages 6-7 were utilized for all experiments involving HBMEC and HREC.”

Both cell types have been specified to maintain their endothelial characteristics up to passage 10. We made every effort to use the lowest possible passage number (6-7) to preserve the endothelial properties of both cell types. Additionally, we stained for the endothelial marker VE-Cadherin at both analysis time points to confirm an intact endothelial phenotype in untreated cells.

5.     “Figure 1: Anaphylatoxins C3a and C5a differentially regulate endothelial monoculture dnCI in a 117 dose-dependent manner. (A, B) Monitoring of dnCI changes in response to 50 nM, 100 nM and 500 118 nM of anaphylatoxins C3a and C5a revealed a transient decrease in dnCI by 500 nM C5a after two 119 hours of treatment in HBMEC. 24 hours after anaphylatoxin stimulation, 500 nM C3a significantly 120 decreased the paracellular resistance in HBMEC……..” For n=6 or 9, were the cells used all from the same passage, or were they from different passages?

For both cell types, we utilized passages 6-7 for all experiments, and we have included this information in the methods section (see point 4). According to the distributors, both HBMEC and HREC remain viable up to passage 10. We recognize that using different passages when working with primary cells is critical; therefore, we ensured to include only experiments with the lowest passage possible and refrained from using cells that are approaching the maximum passage as indicated by the vendors.

  1. “Both anaphylatoxins were diluted in the respec-
    563 tive culture medium. Treatment was administered at concentrations of 50 nM, 100 nM and
    564 500 nM for real time cell analysis (RTCA) and 500 nM for immunocytochemical staining
    565 and qRT-PCR samples. Unaltered culture medium was added to the untreated control. 566”
    What were the criteria for dose selection? Why was only the 500 nM dose used for immunostaining?

    We selected concentrations of 50, 100, and 500 nM for C3a and C5a based on their common usage in experimental studies involving anaphylatoxins (e.g. doi: 10.1007/s00011-010-0178-4, 10.3389/fimmu.2020.615236). Baseline levels typically range from 20-60 nM for C3a and 1-5 nM for C5a, but these concentrations can rise significantly in conditions such as sepsis, trauma, or autoimmune diseases—reaching 200-500 nM for C3a and 10-50 nM for C5a. Importantly, while these are systemic levels, localized complement activation, such as that triggered by autoantibody binding in NMOSD, can lead to a localized surge in anaphylatoxin concentrations, further elevating the local protein levels.

We selected the 500 nM dose of both anaphylatoxins because it produced the most pronounced differences regarding the treatment with 500 nM C3a in brain endothelial cells (HBMEC, Fig. 1A). Specifically, we observed that 500 nM C3a decreased paracellular impedance in brain endothelial cells but not in retinal cells. Consequently, we evaluated how the responses of both endothelial cell types differ concerning the expression of the endothelial marker VE-Cadherin, precursor proteins, and anaphylatoxin receptors.

 We have included the following statement for further clarification:

Lines 133-135: “We observed a tissue-specific effect of 500 nM C3a on paracellular barrier properties of cerebral and retinal endothelial cells (Figure 1), and consequently wanted to investigate whether this effect extends to the adherens junction marker VE-cadherin (CDH5).”

  1. “Monitoring of dnCI changes in response to 50 nM, 100 nM and 500 118 nM of anaphylatoxins C3a and C5a revealed a transient decrease in dnCI by 500 nM C5a after two 119hours of treatment in HBMEC. 24 hours after anaphylatoxin stimulation, 500 nM C3a significantly” Was cell viability assessed in response to the anaphylatoxins? Was an MTT assay or another viability test performed to confirm the effects of 50 nM, 100 nM, and 500 nM treatments?

    We understand the reviewer's concern regarding the assessment of cell viability, but we are very confident that the cells remain viable. This confidence is based on several robust lines of evidence. First, we utilized real-time cell analysis to measure the integrity of the endothelial barrier across all treatment concentrations (50 nM, 100 nM, and 500 nM of C3a or C5a), providing a functional indication that the cells were healthy. Additionally, while we noted a decrease in dnCI in cells treated with 500 nM C3a, further analysis confirmed that the cells were still viable treated with 500 nM C3a and C5a. Specifically, we examined the endothelial-specific marker VE-Cadherin and observed stable CDH5 gene expression and histological data also confirmed the preservation of cell viability throughout the experiment. These findings are also consistent with results from previous studies assessing endothelial response under toxic conditions: 10.1016/j.expneurol.2024.114919. These results collectively provide strong and reliable evidence that the cells are viable.
  2. “Figure 1: Anaphylatoxins C3a and C5a differentially regulate endothelial monoculture dnCI in a 117 dose-dependent manner” Why was the analysis limited to only 24 hours when the xCELLigence assay was run for more than 50 hours? Additionally, was fresh media and C3a replenished during this time, considering their short half-life? Would extending the time points provide more comprehensive insights?

    The analysis was focused on the 2-hour and 24-hour time points because, at this stage, the standard deviation among technical replicates was low, allowing for a clear observation of effects (shown in Supplement Figure 3). However, we chose to present the entire real-time cell analysis to provide a comprehensive overview of the long-term effects of both anaphylatoxins on the respective endothelial barriers. Given the observed standard deviation after 24 hours, extending the time points would not yield more informative insights.

We did not replenish C3a or C5a during the cell cultivation period, as our aim was to investigate both the short- and long-term effects of a single dose of each anaphylatoxin on endothelial barrier integrity. We focused on the acute effects of these anaphylatoxins. This focus aligns with the nature of NMOSD as a relapsing disease characterized by time-restricted surges in systemic autoantibody availability and complement activation.

We fully agree with the reviewer's point regarding peptide degradation. Given that C3a and C5a are likely to degrade within approximately 2 hours, the observed long-term decrease in endothelial barrier integrity is particularly interesting. It suggests that the effect is not solely dependent on an immediate interaction with C3a, but rather points to a more sustained impact that persists even after the peptides have likely degraded.

Moreover, what makes these findings even more intriguing is that, 24 hours after the initial anaphylatoxin treatment, we observed an increase in C3a protein production in HREC cells, as shown in Figure 4B. This indicates that the cells themselves may respond to the initial treatment by upregulating C3a production, adding a further layer of complexity to the understanding of how these peptides influence endothelial integrity over time.

This long-term response highlights the potential for an indirect, lasting impact of C3a and C5a on cell behavior, even after their degradation.

Replenishing both proteins during cultivation would be valuable for future projects aimed at determining the long-term or chronic effects of anaphylatoxic stress on the endothelial barrier.

  1. “Figure 3: Anaphylatoxin treatment caused increased C3 expression in HREC and increased C3AR1 199 expression in HBMEC. (A, B, C, D) qRT-PCR analysis revealed no effect of anaphylatoxin treatment 200 on C3 expression in HBMEC, but disclosed an increase in C3 expression in 500 nM C5a treated HREC 201 2 hours post-treatment and a 500 nM C3a and 500 nM C5a mediated increase in C3 gene expression 202 24 hours post-treatment in HREC. Two hours exposure to 500 nM C5a increased C3AR1 expression…” What is the sample size (n) used in this analysis? In panel B, there appears to be a high standard deviation. Can the authors provide justification for this? Were the data collected from four separate cultures considered technical replicates? What passage number was used for the cells?"

    We have included the sample size for all qPCR data in the methods section:

Lines 670-671: “qPCR data reflects the median of a sample size of n = 4-8 as represented in the corresponding graphs.”

 The standard deviation reflects the use of four separate cultures as technical replicates, leading to interassay variance. By repeating the same experiment four times, we ensured that all observed effects are consistent across different cultivations and are not isolated occurrences. We included all data generated from these four distinct cultivation steps, thus avoiding any potential bias for the sake of achieving a lower standard deviation.

We utilized passages 6 and 7, which may also contribute to the higher standard deviation. We have included the following statement in the Materials & Methods section:

Lines 622-623: “Passages 6-7 were utilized for all experiments involving HBMEC and HREC.”

  1. “Figure 6: C5AR1 transcript expression remains unchanged, but protein detection changes under an- 301aphylatoxic stress. (A, B) qRT-PCR analysis showed no difference in C5AR1 gene expression be- 302 tween untreated and C3a or C5a treated HBMEC and HREC. (C) 500 nM C3a and 500 nM C5a 30….” Why are untreated HREC showing different C5AR1 levels at 2 hours compared to 24 hours? Why does the 2-hour time point show higher expression in both cell types in the untreated control group compared to the 24-hour time point?

C5AR1 transcript levels remain comparable regardless of treatment or duration. We acknowledge that there is a difference in C5aR1 protein signal intensity between 2 and 24 hours in the histological dataset for the untreated cells. At this stage, we have chosen to focus on the comparison between treatments rather than the two time points, and have therefore not addressed this issue further. Based on your comment, we have adjusted the order of the figures and sorted the data by time point rather than cell type. This way, the restructuring facilitates a clearer and more direct comparison between the two cell types.

However, if we were to interpret the data, several potential explanations could be considered for this result:

  1. Endothelial cells may adapt further to cultivation conditions over time, leading to changes in protein arrangement in the cell. This phenomenon has been well-documented, particularly in retinal pigment epithelial cells (e.g. actin filaments, bestrophin). Consequently, C5aR1 protein might diminish with prolonged culture, especially in cells not exposed to stimuli that would maintain or upregulate its expression.
  2. We have demonstrated that endothelial cells express C5, which may lead to the production of C5a. Upon ligand binding (autocrine regulation), C5aR1 can undergo internalization into endosomes, which could reduce its surface expression and result in a weaker signal during surface-directed immunocytochemical staining. Relocation to intracellular compartments may decrease the detectable signal. This is increasingly recognized, as recent studies indicate that the complement system extends beyond its traditional functions and also operates intracellularly.

  1.   “….. 500 nM C3a and 500 nM C5a 303 treatment reduced C5aR1 protein signal in HBMEC after 2 hours of treatment. After 24 hours, the 304 C5aR1 signal decreased in untreated cells but increased in both 500 nM C3a and 500 nM C5a treated 305 HBMEC…..” The representative image does not accurately reflect the stated observations. Better-quality images that support the authors' claims should be included. Additionally, performing quantification using ImageJ would enhance the analysis and understanding of the results. Highlighting just one location in the whole slide may not adequately represent the overall behavior of the cells.

    Thank you for this insightful comment. We completely agree that quantifying the immunocytochemical analyses enhances the manuscript and adds rigor to the study. As a result, we have quantified all stainings presented in Figures 2, 4, 5, 6, and 7. We employed three technical replicates for the analysis and assessed fluorescent intensity, normalizing it to the untreated control, consistent with our other data. We believe that our statements are now well-supported by the quantitative data. Accordingly, we have revised all figure legends and made the necessary adjustments in the results section.

    12.  “E) Immunocytochemistry revealed a highly functional barrier in HBMEC treated with 500 nM C3a 359……..+ 50 nM C5a, 2 hours post-treatment.” A better representative image is required. From Figure 7E, it appears that the combination treatment shows a stronger Cadherin barrier compared to the untreated control group. Can the authors provide justification for this observation? Additionally, how was the combinatorial dose composition (10:1) for the treatment selected?

Thank you for your keen observation. As previously mentioned, we have conducted a quantitative analysis of the immunocytochemical data, and the corresponding results have been included in the figure legends and the results section. Our analysis revealed no significant differences between the untreated cells and those subjected to the combination treatment of C3a and C5a. However, notable differences were observed between C3a-treated and untreated cells, as well as between C3a-treated and C3a+C5a-treated HBMEC cells.

 The combinatorial dose of 10:1 was chosen based on the titration experiments we performed using real-time cell analysis (RTCA) (Figure 7A, B). We observed a barrier-disruptive effect in brain endothelial cells specifically after treatment with 500 nM C3a, which prompted us to titrate C5a against this concentration to explore potential combinatorial effects. Our findings indicated that the 10:1 combination ratio effectively mitigated the barrier-disruptive effect induced by 500 nM C3a alone. Consequently, we applied this combination, identified as the most effective treatment in the RTCA, to our immunocytochemical analysis as well.

Round 2

Reviewer 2 Report

Comments and Suggestions for Authors

Thanks to the Authors for considering my comments. I think this version of the paper is fine and I have no further observations.